# Myonuclear alterations associated with exercise are independent of age in humans

E. Battey[1,2,3] , J. A. Ross[2], A. Hoang[2], D. G. S. Wilson[4], Y. Han[1], Y. Levy[1] , R. D. Pollock[1] ,
M. Kalakoutis[1,5], J. N. Pugh[6] , G. L. Close[6], G. M. Ellison-Hughes[1], N. R. Lazarus[1], T. Iskratsch[4,5],
S. D. R. Harridge[1] , J. Ochala[1,7] and M. J. Stroud[2]

[1]*Centre for Human & Applied Physiological Sciences, Faculty of Life Sciences & Medicine, King's College London, London, UK*
[2]*British Heart Foundation Centre of Research Excellence, School of Cardiovascular Medicine and Sciences, King's College London, London, UK*
[3]*Novo Nordisk Foundation Center for Basic Metabolic Research, University of Copenhagen, Copenhagen, Denmark*
[4]*School of Engineering and Materials Science, Queen Mary University of London, London, UK*
[5]*Randall Centre for Cell and Molecular Biophysics, Faculty of Life Sciences & Medicine, King's College London, London, UK*
[6]*School of Sport and Exercise Sciences, Liverpool John Moores University, Liverpool, UK*
[7]*Department of Biomedical Sciences, Faculty of Health and Medical Sciences, University of Copenhagen, Copenhagen, Denmark*

Handling Editors: Michael Hogan & Kevin Murach

The peer review history is available in the Supporting Information section of this article
(https://doi.org/10.1113/JP284128#support-information-section).

This article was first published as a preprint. Battey E, Ross JA, Hoang A, Wilson DGS, Han Y, Levy Y, Pollock RD, Kalakoutis M, Pugh JN, Close GL, Ellison-Hughes GM, Lazarus NR, Iskratsch T, Harridge SDR, Ochala J, Stroud, MJ. 2022. Myonuclear alterations associated with exercise are independent of age in humans. bioRxiv. https://doi.org/10.1101/2022.09.20.506578

The Journal of Physiology

**Abstract** Age-related decline in skeletal muscle structure and function can be mitigated by regular exercise. However, the precise mechanisms that govern this are not fully understood. The nucleus plays an active role in translating forces into biochemical signals (mechanotransduction), with the nuclear lamina protein lamin A regulating nuclear shape, nuclear mechanics and ultimately gene expression. Defective lamin A expression causes muscle pathologies and premature ageing syndromes, but the roles of nuclear structure and function in physiological ageing and in exercise adaptations remain obscure. Here, we isolated single muscle fibres and carried out detailed morphological and functional analyses on myonuclei from young and older exercise-trained individuals. Strikingly, myonuclei from trained individuals were more spherical, less deformable, and contained a thicker nuclear lamina than those from untrained individuals. Complementary to this, exercise resulted in increased levels of lamin A and increased myonuclear stiffness in mice. We conclude that exercise is associated with myonuclear remodelling, independently of age, which may contribute to the preservative effects of exercise on muscle function throughout the lifespan.

(Received 17 November 2022; accepted after revision 19 December 2022; first published online 4 January 2023)

**Corresponding authors** J. Ochala: Department of Biomedical Sciences, Faculty of Health and Medical Sciences, University of Copenhagen, Copenhagen, Denmark. Email: julien.ochala@sund.ku.dk; M. J. Stroud: British Heart Foundation Centre of Research Excellence, School of Cardiovascular Medicine and Sciences, King's College London, London SE5 9NU, UK. Email: matthew.stroud@kcl.ac.uk

**Abstract figure legend** Structural and mechanical properties of myonuclei in trained young and aged individuals. In skeletal muscle fibres from trained individuals, myonuclei are more spherical, have greater lamin A and are stiffer compared to untrained counterparts. This may protect nuclei from damage when subjected to contractile forces during exercise, and permit effective transduction of these forces to regulate gene expression and signalling pathways (mechano-transduction). In skeletal muscle from untrained older individuals, myonuclei are more elongated, nuclear lamina levels are lower and myonuclei are more deformable. This may increase susceptibility to myonuclear damage and defective mechanotransduction, contributing to declines in muscle mass and function.

## Key points

- The nucleus plays an active role in translating forces into biochemical signals.
- Myonuclear aberrations in a group of muscular dystrophies called laminopathies suggest that the shape and mechanical properties of myonuclei are important for maintaining muscle function.
- Here, striking differences are presented in myonuclear shape and mechanics associated with exercise, in both young and old humans.
- Myonuclei from trained individuals were more spherical, less deformable and contained a thicker nuclear lamina than untrained individuals.
- It is concluded that exercise is associated with age-independent myonuclear remodelling, which may help to maintain muscle function throughout the lifespan.

## Introduction

Human lifespan has increased substantially over the past half-century and this trend is projected to continue (UN Department of Economic and Social Affairs, 2022). However, this has not been accompanied by an equivalent extension of the healthspan in old age; instead, morbidity has been extended, and independence and quality of life

**Ed Battey** completed his bachelor's degree in Sport and Exercise Science at Loughborough University followed by an MSc in Human & Applied Physiology at King's College London, in which he undertook a research project in Professor Simon Hughes's lab studying the role of a transmembrane muscle cell fusion protein in zebrafish. He then completed an MRC DTP PhD project supervised by Dr Matthew Stroud and Dr Julien Ochala. This project involved studying the potential roles of myonuclear structure and function in exercise and ageing, leading to the work published in this paper. Recently, he completed an MRC-funded project in Professor Kei Sakamoto's lab and will next commence a post-doc researching Rho-GTPase signalling in skeletal muscle in Dr Lykke Sylow's lab at the University of Copenhagen.
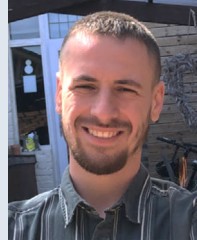

attenuated (Brown, 2015). Thus, a 'managed compression of morbidity' is essential to address social and economic issues associated with an extended lifespan (Brown, 2015). A contributing factor to morbidity is the decline in skeletal muscle structure and function associated with ageing. Muscle contractions produce force, allowing us to carry out whole-body movements such as walking, stair-climbing or rising from a chair – movements essential for independence and quality of life. The ageing process, however, is compounded by the physically inactive status of individuals in a technologically advanced society (Guthold et al., 2018; Nikitara et al., 2021). Furthermore, physical inactivity can accelerate the decline in physiological function that inevitably occurs during later years of life (Lazarus & Harridge, 2017; Shur et al., 2021).

Skeletal muscle structure and function can be better maintained in old age by exercise, but the mechanisms behind this remain poorly understood (Lazarus et al., 2019; Pollock et al., 2015; Wroblewski et al., 2011). An area of research which is understudied is how exercise and ageing influence the ability of skeletal muscle to translate force into biochemical signals (mechanotransduction) at the subcellular level. This is pertinent given the contractile nature of muscle and the opposing effects of exercise and inactivity on the frequency and intensity of muscle contractions. Emerging data suggest that the nucleus is a critical mechanosensor that orchestrates cell structure, function and adaptive responses (Kirby & Lammerding, 2018). Indeed, the shape and mechanical properties of nuclei appear to regulate gene expression by altering genome organisation and ultimately influencing broader transcriptional profiles (Kalukula et al., 2022; Kirby & Lammerding, 2018; Piccus & Brayson, 2020; Tajik et al., 2016). Additionally, altered nuclear shape and nuclear envelope stretching can expand nuclear pore complexes and ion channels, facilitating translocation, respectively, of mechanosensitive transcription factors Yes-associated protein/transcriptional coactivator with PDZ-binding motif (Yap/Taz) and myocardin-related transcription factor (MRTF-A) or ions such as $Ca^{2+}$, altering gene expression and signalling (Kirby & Lammerding, 2018; Maurer & Lammerding, 2019; Ross & Stroud, 2021; Shen et al., 2022).

Abnormal nuclear structure and responses to forces are hallmarks of numerous diseases that result in skeletal muscle weakness and premature ageing, commonly caused by mutations in nuclear envelope and associated proteins (Battey et al., 2020; Earle et al., 2020; Goldman et al., 2004; Kalukula et al., 2022; Ross et al., 2019). Such proteins physically link the nucleus to the cytoskeleton, providing a nexus for mechanotransduction (Banerjee et al., 2014; Crisp et al., 2006; Kirby & Lammerding, 2018; Ross & Stroud, 2021). The nuclear lamina, which lines the inner nuclear membrane and comprises lamins A/C, B1 and B2, tethers chromatin to the nuclear periphery, associates with nuclear pore complexes, and connects the nucleoskeleton to the cytoskeleton via linker of nucleoskeleton and cytoskeleton (LINC) complex (Osmanagic-Myers et al., 2015; Owens et al., 2021; Stroud, 2018; Stroud et al., 2017). Within this prominent location, the nuclear lamina is critically positioned to sense cytoskeletal forces to regulate gene expression, biochemical signalling and overall cell function and adaptation (Cho et al., 2017; Maurer & Lammerding, 2019).

Importantly, various diseases caused by mutations in genes encoding nuclear lamina proteins (termed laminopathies) primarily affect mechanically active muscle tissue and result in aberrant nuclear shape, structural integrity and mechanotransduction (Earle et al., 2020; Janin et al., 2017; Shin & Worman, 2021). One such laminopathy is Hutchinson–Gilford progerin syndrome, a premature ageing syndrome caused by a mutation in the gene encoding lamin A/C (Goldman et al., 2004; Merideth et al., 2008). Defective lamin A/C expression also results in muscular dystrophy characterised by muscle weakness (such as autosomal dominant Emery–Dreifuss muscular dystrophy, limb-girdle muscular dystrophy type1B, and *Lmna*-congenital muscular dystrophy) (Bonne et al., 1999; Maggi et al., 2016). Thus, myonuclear shape, nuclear envelope proteins and nuclear mechanics are dysregulated in muscle pathologies and premature ageing syndromes and may have important roles in age-related muscle dysfunction.

Lamin A/C has been shown to be required for normal nuclear mechanics in myotubes and for cardiac and skeletal muscle overload hypertrophy responses in mice (Cupesi et al., 2010; Earle et al., 2020; Owens et al., 2021). Indeed, a congenital mutation in lamin A/C causing muscular dystrophy resulted in altered nuclear mechanics, attenuated hypertrophy reduced force capacity in response to functional overload in mouse skeletal muscle (Owens et al., 2021). In cardiac tissue, in response to pressure overload, haploinsufficient lamin A/C mice demonstrated reduced ventricular mass and myocyte size and impaired mechanotransduction (Gerhart-Hines et al., 2007; Gurd, 2011; Little et al., 2010). Collectively, these studies hint at a potential lamin A/C-dependent mechanosensitive signalling cascade in regulating both muscle hypertrophy and oxidative exercise adaptations.

Despite the large amount of evidence suggesting the importance of nuclear shape, mechanics and lamina in premature ageing and muscle pathologies, their roles in normal ageing and exercise are poorly understood (Cupesi et al., 2010; Earle et al., 2020; Gerhart-Hines et al., 2007; Gurd, 2011; Little et al., 2010; Owens et al., 2021). To this end, we investigated whether ageing and exercise affected structure and function of skeletal muscle nuclei. Single muscle fibres from young and older trained and untrained individuals were isolated and myonuclear

structure and function analysed. Detailed 2D and 3D morphological analyses of myonuclei revealed striking nuclear shape differences in trained individuals compared to untrained individuals, regardless of age. Additionally, myonuclei from trained individuals had increased nuclear lamina deposition and were less deformable compared to untrained counterparts. Consistently, skeletal muscle from trained mice had increased levels of lamin A and increased nuclear stiffness. Our data suggest for the first time in humans that exercise is associated with differences in myonuclear shape and mechanics, which likely mitigate the deleterious effects of inactive ageing.

## Methods

### Ethical approval

Prior to participation, written informed consent was obtained from all subjects. Procedures were approved by the Fulham Research Ethics Committee in London (12/LO/0457), Westminster Ethics Committee in London (12/LO/0457) or Liverpool John Moores ethics committee (H17SPS012) and conformed to the standards set by the *Declaration of Helsinki*. All human tissues were collected, stored and analysed in accordance with the Human Tissue Act. Procedures were performed in accordance with the Animals (Scientific Procedures) Act 1986 (UK Home Office); King's College London License number: X24D82DFF, ethics code: PDB33C80B.

### Participant characteristics and ethics

Four mixed sex groups were recruited to participate in the current study ($n = 6$ per group). These groups were: younger untrained healthy (YU) ($33 \pm 9.5$ years), younger trained marathon runners (YT) ($32 \pm 5.4$ years), older untrained individuals (OU) ($79 \pm 11.3$ years) and older highly trained cyclists (OT) ($75.5 \pm 3.2$ years) (Table 1). The YU group was considered healthy, but not necessarily sedentary, as two of the participants had been participating in low-level recreational sport activities (<2 sessions/week) at the time of the study. Thus, the young cohort consisted of low-level physically active and sedentary individuals. Participants were considered healthy if they met the criteria outlined by Greig et al. (1994). The exclusion criteria from a healthy classification were smoking or consuming alcohol excessively, known hypertension or other cardiovascular, musculoskeletal or neurological conditions, or if they were on any medication (acute or chronic). The YT group consisted of trained marathon runners ($\dot{V}_{O_2peak}$ $56.7 \pm 6.6$ ml/kg/min, mean $\pm$ SD). In the YT group, the fastest running times (mean $\pm$ SD) in the previous 18 months over marathon, half marathon and 5 km distances were $204.5 \pm 14.2$ min,

$88.5 \pm 3.3$ min and $19.8 \pm 1.3$ min, respectively. The OU group, used as a model for muscle disuse in old age, was a previously characterised cohort who underwent dynamic hip screw insertion surgery. The patients completed a basic physical health questionnaire and were considered eligible if they did not suffer from neuromuscular disease, although some had underlying health conditions (Table 1). The OT group consisted of previously characterised individuals (Pollock et al., 2015) who were amateur master endurance cyclists. Master cyclists were included if they were able to cycle 100 km in under 6.5 h (males) or 60 km in under 5.5 h (females). Participants must have had completed this distance within the specified time on two occasions in the 3 weeks prior to the date of participation in the study.

### Obtaining and processing skeletal muscle samples and isolating single muscle fibres

Vastus lateralis samples were obtained as previously described (Pollock et al., 2018). Approximately 60 mg of the biopsy sample was then placed in relaxing solution (mM: 77.63 KCL, 10 imidazole, 2 MgCl$_2$, 2 EGTA, 4.05 ATP in distilled water, pH 7.0) in a Petri dish on ice. Following excision, muscle samples (submerged in relaxing solution in a Petri dish) were divided into bundles of approximately 100 muscle fibres using forceps under a stereo microscope (Zeiss Microscopy, Jena, Germany, Stemi 2000-C) with a separate light source (Zeiss Stereo CL 1500 ECO). The ends of the bundles were then tied onto glass capillary tubes using surgical silk (LOOK SP102) and stretched to approximately 110% of the original length. These bundles were subsequently placed into 1.5 ml Eppendorf tubes, containing skinning solution (relaxing solution with 50% (v/v) glycerol), at 4°C for 48 h to permeabilise the muscle fibres by disrupting the lipid bilayer of the sarcolemma, leaving myofilaments, intermediate filaments and nuclear envelope intact (Frontera & Larsson, 1997; Konigsberg et al., 1975; Stienen, 2000; Wood et al., 1975). Samples were then treated in ascending gradients of sucrose dissolved in relaxing solution (0.5 M, 1 M, 1.5 M, 2 M) for 30 min to prevent cryodamage (Frontera & Larsson, 1997). In a Petri dish containing 2 M sucrose, fibres were then removed from the glass capillary tubes before being placed in cryovials and snap-frozen in liquid nitrogen.

For immunofluorescence and nuclear mechanics experiments, muscle fibre bundles were placed in descending concentrations of sucrose dissolved in relaxing solution, for 30 min in each solution (2 M, 1.5 M, 1 M, 0.5 M, 0 M). Samples were transferred to skinning solution at −20°C until the day of an experiment. To isolate single muscle fibres, muscle bundles were placed in skinning solution in a Petri dish on an ice block. One

**Table 1. Participant characteristics**

| Age | Sex | Height (cm) | Weight (kg) | BMI (kg/m$^2$) | Comment | Drug intake |
|---|---|---|---|---|---|---|
| Younger untrained | | | | | | |
| 25 | F | 164 | 55 | 20.4 | | |
| 27 | M | 184 | 80 | 23.6 | | |
| 22 | M | 178 | 75 | 23.7 | | |
| 42 | F | 172 | 73 | 24.7 | | |
| 38 | F | 158 | 48 | 19.2 | | |
| 44 | F | 166 | 52 | 18.9 | | |
| Young marathon runners (younger trained) | | | | | | |
| 35 | M | 182 | 71 | 21.4 | | |
| 32 | M | 176 | 67 | 21.6 | | |
| 32 | F | 170 | 64 | 22.1 | | |
| 22 | M | 190 | 70 | 19.4 | | |
| 38 | M | 170 | 68 | 23.5 | | |
| 34 | F | 160 | 55 | 21.5 | | |
| Older untrained | | | | | | |
| 59 | M | 164 | 53 | 19.7 | Osteoporosis | Perindopril, amlodipine, etidronate, adcal |
| 91 | F | | | | Osteoporosis, diverticulitis – Hartmann's bowel operation | Furosemide, paracetamol, colecalciferol, docusate |
| 77 | M | | | | Osteoporosis | Paracetamol, tramadol, omeprazol, renetadine, mirapixin, procloperizone |
| 82 | F | 152 | 57 | 24.7 | Osteoporosis, hypertension, rheumatoid arthritis, type 2 diabetes | Metformin, denusomab, levothyroxine, allopurinol |
| 88 | F | 156 | 64 | 26.3 | Osteoporosis, breast cancer, left mastectomy in remission | Levothroxine, atenolol |
| 77 | F | | 61 | | Osteoporosis, rheumatoid arthritis | Sulfasalazine |
| Master cyclists (older trained) | | | | | | |
| 71 | M | 180 | 83 | 25.7 | | |
| 76 | F | 157 | 58 | 23.4 | | |
| 79 | F | 159 | 53 | 21.1 | | |
| 75 | M | 172 | 62 | 20.8 | | |
| 73 | M | 170 | 70 | 24.0 | | |
| 79 | F | | | | | |

end of the muscle bundle was held using extrafine forceps, whilst single fibres were pulled from the end of the bundle. During this process, care was taken to restrict contact to the ends of fibres as much as possible to avoid damage. To normalise muscle fibre tension and orientation, muscle fibres were mounted on a half-split grid for transmission electron microscopy glued to a coverslip (Levy et al., 2018; Ross et al., 2017). Fibres were then immunostained and imaged or analysed by a nanoindenter to assess nuclear mechanics.

### Immunostaining, imaging and analysis of single muscle fibres

The first steps of each staining protocol were fixing in 4% paraformaldehyde for 15 min and permeabilising in 0.2% Triton X-100 for 10 min. When primary antibodies were used, fibres were blocked using 10% goat serum (Sigma-Aldrich, Gillingham, UK, G9023) in phosphate-buffered saline (PBS) for 1 h at room temperature before incubation in primary antibody solution overnight at 4°C. Muscle fibres were then incubated in a solution containing direct stains and secondary antibodies for 1 h (see list of primary and secondary antibodies in Table 2). Finally, fibres were mounted in Fluoromount-G® (Thermo-Fisher Scientific) or DAKO (Agilent, Santa Clara, CA, USA) mounting medium. Between each step of staining protocols, fibres were washed 4 times in PBS.

For two-dimensional analysis of myonuclear shape, single plane or Z-stack images (1 $\mu$m Z increments) were acquired using a ×40 air objective and a Zeiss Axiovert 200 microscope system. Two-dimensional myonuclear

**Table 2. Antibodies and direct stains used in immunofluorescence experiments**

| Antibody/stain | Concentration | Company/Lab | Catalogue number | Species |
|---|---|---|---|---|
| Lamin A | 1:500 | Sigma-Aldrich | L1293 | Rabbit |
| Nesprin-1 (8C3) | 1:400 | Glen Morris lab | — | Rabbit |
| MYH7 (A4.951) | 1:50 | Santa Cruz Biotechnology | sc-53090 | Mouse |
| $\alpha$-Actinin (EA-53) | 1:500 | Sigma-Aldrich | A7811 | Mouse |
| Alexa Fluor™ 594 Phalloidin | 1:100 | Thermo Fisher Scientific | A12381 | — |
| Alexa Fluor™ 488 Goat anti-Rabbit IgG H+L | 1:800 | Thermo Fisher Scientific | A-11008 | Goat |
| Alexa Fluor™ 555 Goat anti-Mouse IgG H+L Superclonal | 1:800 | Jackson Immunoresearch | 715-165-150 | Goat |
| Alexa Fluor™ 647 Goat anti-Mouse IgG H+L | 1:800 | Thermo Fisher Scientific | A-21235 | Goat |
| Hoechst 33342 | 1:800 | Thermo Fisher Scientific | H3570 | — |
| DAPI | 1:800 | Thermo Fisher Scientific | D3571 | — |

Jackson ImmunoResearch Laboratories, West Grove, PA, USA; Santa Cruz Biotechnology, Dallas, TX, USA; Sigma Aldrich, UK; Thermo Fisher Scientific, Waltham, MA, USA. DAPI, 4′,6-diamidino-2-phenylindole.

shape parameters (nuclear area and aspect ratio) were quantified using Fiji software, as previously described (Battey et al., 2022; Schindelin et al., 2012). Single plane or maximum intensity projection images were processed with a rolling ball background subtraction (150 pixels), Gaussian blur filter (2 pixels radius) and despeckle function before thresholding the 4′,6-diamidino-2-phenylindole (DAPI) signal (initially with 'Otsu dark' setting then adjusting as necessary). Laterally located nuclei (i.e. positioned around the sides of muscle fibres) were excluded from analysis as nuclei in this position are orientated perpendicular to, rather than facing, the objective lens. Sarcomere length was quantified by measuring the distance between ten sarcomeres using the segmented line tool in Fiji and dividing this value by 10.

For three-dimensional analysis of myonuclear shape and lamin A organisation, Z-stack images (0.2 $\mu$m Z increments) were acquired using a ×60 oil objective and a Nikon (Tokyo, Japan) spinning disk confocal microscope system. To quantify three-dimensional shape parameters (sphericity; skeletal length/diameter, referred to as 3D aspect ratio), the DAPI signal was thresholded and analysed using Volocity software (Perkin Elmer, Waltham, MA, USA). Skeletal length is the maximum length of the object, which is eroded evenly from its border inwards until it consists of a 1-voxel-thick skeletal representation along its entire length. Skeletal diameter is the diameter of a cylinder if it had a length equal to the skeletal length of the object and a volume equal to the object's measured volume (from Volocity User Guide). Representative images were produced by generating standard deviation pixel projections of Z-stacks in Fiji.

To visualise and analyse lamin A at super-resolution level, Z-stack images (0.1 $\mu$m increments) were acquired with a ×100 oil objective (numerical aperture 1.5) and a Nikon instant Structured Illumination Microscope (iSIM)

system. At least six fibres were imaged per individual, with each image including one to seven myonuclei. To improve contrast and resolution (by two-fold compared to confocal microscopy), iSIM images were deconvolved using inbuilt algorithms in Nikon Elements software (3D Blind algorithm with 15 iterations and spherical aberration correction) (Curd et al., 2015; York et al., 2013). The organisation of the nuclear lamina was analysed using Fiji software. Line scan analysis of lamin A staining was performed by using the plot profile tool. Nuclear lamina deposition (arbitrary units) was quantified by using a full-width at half-maximum macro to fit a Gaussian curve to pixel intensity profiles of lamin A stains. Measurements using the tool were taken in mid-focal planes, with an average taken from a minimum of six measurements per nuclei. For nuclear invagination length analysis, a Fiji plugin called Ridge Detection was used (Steger, 1998).

## Assessment of myonuclear mechanics in single muscle fibres

To assess myonuclear mechanics in single muscle fibres, the extent of nuclear deformation with increasing fibre tension was quantified (Chapman et al., 2014; Shah & Lieber, 2003). Muscle fibres from OT and OU were stretched to different tensions, fixed, and stained with Hoechst and $\alpha$-actinin to visualise myonuclei and Z-discs of sarcomeres, respectively. Images were acquired as before for 2D analysis, and sarcomere length was correlated with nuclear aspect ratio, to assess myonuclear shape at various extents of muscle fibre stretch.

To determine stiffness, elastic and viscosity properties of myonuclei, nanoindentation was carried out on mounted single muscle fibres. Experiments were performed using an Optics11 Chiaro nanoindenter attached to a Leica Microsystems (Wetzlar, Germany)

**Table 3. Dynamic mechanical analysis parameters for nanoindentation**

| Frequency (Hz) | Indentation depth (nm) | Periods | Time (s) |
|---|---|---|---|
| 1 | 500 | 5 | 2 |
| 2 | 500 | 5 | 2 |
| 4 | 500 | 20 | 2 |
| 10 | 500 | 20 | 2 |

DMI-8 inverted microscope. Mounted muscle fibres were stained with Hoescht to locate myonuclei, and only nuclei at the nearest surface of the muscle fibre to the indenter were analysed. Nanoindentation was performed with a 9-$\mu$m-diameter spherical probe (2.1 $\mu$m contact radius at 0.5 $\mu$m indentation depth, corresponding to approximately half the nuclear radius, whereby an indentation at 0.5 $\mu$m depth measures primarily the nuclear *vs.* the cytoskeletal contribution to the stiffness; Guerrero et al., 2019). Approach and retraction speeds were set to 500 nm/s. The Hertzian contact model was used to fit the load-indentation data for calculation of Young's modulus.

Dynamic Mechanical Analysis (DMA), which uses a cyclic motion with frequency while controlling displacement or load, was used to calculate the frequency-dependent storage modulus ($'''$), loss modulus ($''''$), and the dissipation factor tan delta ($'''/''''$) of myonuclei, corresponding to the elastic properties of nuclei; 1, 2, 4, and 10 Hz frequencies were used (Table 3).

### Mouse high intensity interval training programme

**Animal handling.** Twenty-four male 10-week-old C57BL/6 mice were maintained in groups of four in cages, lined with wood shavings, cardboard rolls and cleaned weekly, in an automated room for photoperiod control (light–dark cycle 12 h/12 h). Animals were provided with water and a standard chow diet *ad libitum*. Twelve mice were trained on a treadmill over 8 weeks, with a complementary sedentary group left in cages for an equivalent time-period. At the end of the 8 weeks, mice were sacrificed by cervical dislocation and tibialis anterior muscle was excised from both legs of each mouse. Muscle from one leg was placed in skinning solution before cryopreservation and storage at −80°C for later analysis through nanoindentation. Muscle from the contralateral leg was snap-frozen in liquid nitrogen for western blot analysis.

**Treadmill familiarisation, determination of peak running velocity and training programme.** Mice were familiarised

to the treadmill with 5 days of low intensity running (5 cm/s on the first day, increasing the speed by 5 cm/s on the second, third and fifth day, to end with a speed of 20 cm/s). Treadmill incline was set at 0° on day 1, 5° on days 2 and 3, and 10° on day 5. Peak running velocity ($V_{Peak}$) was determined to estimate maximal aerobic capacity and allow standardisation of the intensity of running during the training programme. Exercise prescription based on $V_{Peak}$ and $\dot{V}_{O_2max}$ results in similar aerobic adaptations in both humans and mice (Manoel et al., 2017; Picoli et al., 2018). $V_{Peak}$ was determined based on the method outlined by Picoli et al. (2018), adapted to incorporate a ramp, rather than incremental, increase in running speed, as suggested by Ayachi et al. (2016). Testing commenced with a warm-up for 4 min at 10 cm/s, before increasing the speed gradually to 19 cm/s over the next minute (approximately 1 cm/s every 6.5 s). Running speed was then increased by 1 cm/s every 20 s until exhaustion, characterised by incapacity to keep running for more than 5 s (Mille-Hamard et al., 2012).

Mice ran four times per week for 8 weeks, based on a programme that showed ∼50% improvement in $\dot{V}_{O_2max}$ (Høydal et al., 2007; Kemi et al., 2002). After a warm-up at 10 cm/s for 6 min, mice ran at approximately 80–90% $V_{Peak}$ for three bouts of 8 min intermitted by 2 min active recovery at 50–60% $V_{Peak}$. Table 4 shows the speed and inclines of the training programme. Muscle was excised from mice (sacrificed by cervical dislocation) 72 h after the final exercise session to exclude potential confounding effects observed acutely after exercise (Carmichael et al., 2005; Neubauer et al., 2014).

### Quantification of mouse skeletal muscle protein content by western blotting

Tissue samples were lysed in 5 M urea, 2 M thiourea, 3% SDS, 75 mM dithiothreitol, 0.03% Bromophenol Blue, and 0.05 M Tris–HCl, and homogenised in a Precellys (VWR) 24 tissue homogeniser machine kept at 4°C. Samples were then sonicated for additional homogenisation and shearing of DNA. For western blotting, frozen tissue lysates were thawed and proteins linearised in a heating block at 95°C for 8 min, before loading on 4% to 12% Bis–Tris gels. Proteins were transferred onto nitrocellulose membranes and blocked in 5% non-fat milk powder in Tris-buffered saline with 0.1% Tween 20 (TBS-T) at 4°C for 1 h. The membranes were then incubated with primary antibodies overnight at 4°C, washed in TBS-T 4 times, and subsequently incubated with secondary antibodies for 1 h. After four washes, membranes imaged using a LI-COR Biosciences (Lincoln, NE, USA) Odyssey® CLx imaging system. Antibodies used for western blotting are outlined in Table 5.

**Table 4. Speed and incline of treadmill throughout mouse high intensity interval training programme**

| | | Speed (cm/s) | | | Incline (°) |
|---|---|---|---|---|---|
| | | Group 1 | Group 2 | Group 3 | |
| Week 1 | Exercise | 32, 33 | 29, 30 | 28, 29 | 5 |
| | Active recovery | 19 | 18 | 17 | |
| Week 2 | Exercise | 34, 35 | 31, 32 | 30, 31 | |
| | Active recovery | 19 | 18 | 17 | |
| Week 3 | Exercise | 36, 37 | 33, 34 | 32, 33 | |
| | Active recovery | 20 | 19 | 18 | |
| Week 4 | Exercise | 38, 39 | 35, 36 | 34, 35 | |
| | Active recovery | 20 | 19 | 18 | |
| Week 5 | Exercise | 40, 41 | 37, 38 | 36, 37 | |
| | Active recovery | 21 | 20 | 19 | |
| Week 6 | Exercise | 40, 41, 43 | 38, 39, 40, 40 | 37, 38 | 10 |
| | Active recovery | 21 | 20 | 19 | |
| Week 7 | Exercise | 44, 45, 46, 47 | 41, 42, 43, 44 | 40, 41, 42, 43 | |
| | Active recovery | 22 | 21 | 20 | |
| Week 8 | Exercise | 48 | 46 | 45 | |
| | Active recovery | 23 | 22 | 21 | |

**Table 5. Antibodies used for western blotting**

| Antibody | Company/Lab | Catalogue Number | Species |
|---|---|---|---|
| Nesprin-1 (8C3) | Glen Morris Lab | — | Mouse |
| Nesprin-2 | Didier Hodzic Lab | — | Rabbit |
| SUN1 | Abcam | ab103021 | Rabbit |
| SUN2 | Didier Hodzic Lab | — | Rabbit |
| Lamin A/C | Larry Gerace Lab | — | Rabbit |
| Lamin B1 | Larry Gerace Lab | — | Rabbit |
| Lamin B2 | Abcam | ab151735 | Rabbit |
| Emerin | Santa Cruz Biotechnology | FL-254 | Rabbit |
| Histone H3 (D1H2) | Cell Signaling Technology | 4499T | Rabbit |
| IRDye® 680RD Donkey anti-Mouse IgG secondary antibody | LI-COR Biosciences | 926–68 072 | Donkey |
| IRDye® 800CW Donkey anti-Rabbit IgG secondary antibody | LI-COR Biosciences | 926–32 213 | Donkey |

Abcam, Cambridge, UK; Cell Signaling Technology, Danvers, MA, USA; LI-COR Biosciences, Lincoln, NE, USA; Santa Cruz Biotechnology, Dallas, TX, USA.

## Statistics

Based on an expected mean difference of 15% between young and elderly individuals, an effect size of 1.8, $\alpha = 0.05$ and power $(1 - \beta) = 0.8$, the required sample size was determined as six individuals per group. This study was powered based on nuclear aspect ratio as the primary end-point measurement. Because there were no studies on nuclear shape changes in human muscle fibres, the study was powered based on data showing increased aspect ratio of this magnitude in muscle fibres from mice expressing mutant lamins compared to control mice (Earle et al., 2020). To analyse whether an overall significant difference was present between muscle fibres from the different human groups, two-way analysis of variance (ANOVA) was carried out, followed by a *post hoc* Tukey test to specify which myonuclei or muscle fibres group differences existed. Mean values for each individual were used for two-way ANOVA. For correlation analyses, a simple linear regression was performed to test if slopes were significantly non-zero and non-linear straight-line regression analyses were performed to compare the

slopes of different conditions. A two tailed Student's *t*-test was carried out to analyse differences in protein concentrations and Young's modulus between myonuclei from exercise trained and untrained mice. Mean values for each mouse were used for *t*-tests. With categorical data, individual measurements (myonuclei or muscle fibres) and mean values calculated from these measurements were plotted in the same graph, using SuperPlots (Lord et al., 2020). Individual measurements were plotted as smaller grey points and overall means of each individual or mouse are plotted as larger coloured points. For all statistical tests, $P < 0.05$ indicated significance and $P < 0.07$ was taken to indicate a trend (*$P < 0.05$, **$P < 0.01$, ***$P < 0.001$). Data are presented as means $\pm$ SD except for means $\pm$ SEM in Fig. 5. All data were statistically analysed using Prism 9 (GraphPad Software Inc., San Diego, CA, USA).

### Precision and reproducibility of methods

The same image processing and initial thresholding parameters were used for each image for quantification of nuclear shape, nuclear organisation, lamina deposition and nuclear envelope protein organisation. To quantify the precision and accuracy of the methods used, repeat measurements were taken and the standard deviation and coefficient of variation were calculated. Three repeat measurements of three nuclei were carried out for nuclear aspect ratio and nuclear area. The mean of the standard deviation and coefficient of variation values were then calculated to give a single standard deviation and coefficient of variation value for nuclear aspect ratio and nuclear area. Five repeat measurements of sarcomere length were taken, and five repeat full width at half-maximum measurements of the nuclear lamina were carried out for assessment of the precision of lamina deposition. The standard deviation and coefficient of variation values were 0.02 and 0.67 for nuclear aspect ratio, 1.19 and 1.46 for nuclear area, 0.07 and 0.28 for sarcomere length, and 0.004 and 1.53 for lamina deposition, respectively.

## Results

### Myonuclei in both younger and older trained individuals are more spherical compared to untrained counterparts

Nuclei from laminopathy patients with premature ageing and muscle dysfunction are known to be ruffled and elongated (Earle et al., 2020; Goldman et al., 2004; Park et al., 2009; Tan et al., 2015). Here, we hypothesised that myonuclei would show similar abnormalities in physiological ageing. To investigate the effects of age and exercise

training on myonuclear shape, muscle fibres were isolated from vastus lateralis biopsies taken from younger untrained (YU, 33 $\pm$ 9.5 years), younger trained (YT, 32 $\pm$ 5.4 years), older untrained (OU, 79 $\pm$ 11.3 years) and older trained (OT, 76 $\pm$ 3 years old) individuals.

In contrast to our hypothesis, myonuclei were strikingly rounder in shape in both younger and older trained individuals (Fig. 1A), consistent with recent reports in rodents (Murach et al., 2020; Rader & Baker, 2022). Indeed, the aspect ratio of myonuclei from trained individuals was ∼27–29% lower than untrained counterparts, demonstrating significant differences in roundness (YU, 2.4 $\pm$ 0.3; OU, 2.3 $\pm$ 0.3; YT, 1.7 $\pm$ 0.1; OT, 1.6 $\pm$ 0.2; Fig. 1B). To control for possible differences in fibre tension that may confound interpretation, we normalised myonuclear aspect ratio to sarcomere length and importantly found no differences associated with sarcomere length (Fig. 1C). Sarcomere length was 2.0 $\pm$ 0.2, 2.1 $\pm$ 0.2, 2.0 $\pm$ 0.07 and 2.2 $\pm$ 0.4 $\mu$m in YU, OU, YT and OT, respectively. As further controls, exercise-dependent alterations to myonuclear shape remained apparent in slow muscle fibres expressing myosin heavy chain 7, indicating that fibre-type differences between groups did not influence the effects observed (Fig. 1D). In contrast to changes in nuclear aspect ratio, nuclear area was comparable in muscle fibres from trained and untrained individuals, highlighting that differences to myonuclear aspect ratio were driven by in myonuclear shape rather than size (Fig. 1E).

Next, we performed 3D shape analysis of myonuclei by acquiring serial optical z-slices through whole muscle fibres (Fig. 2A and B). In line with 2D shape changes observed, myonuclei from OT displayed a significant reduction in 3D aspect ratio compared to OU, with a trending reduction observed in YT compared to YU ($P < 0.07$) (Fig. 2C). Furthermore, sphericity values were higher in trained individuals compared to age-matched untrained counterparts, indicating nuclei were more spherical in these groups (Fig. 2D). Importantly, nuclear volumes were comparable across groups, consistent with our 2D analyses (Fig. 2E).

Taken together, 2D and 3D analyses of myonuclear shape revealed striking morphological differences in younger and older trained individuals compared to untrained counterparts.

### Nuclear lamina deposition is greater in skeletal muscle fibres from trained individuals

Lamin A localisation and levels regulate nuclear stiffness and nuclear roundness (Earle et al., 2020; Lammerding et al., 2006; Srivastava et al., 2021; Swift et al., 2013). Additionally, it has recently been shown that a *Lmna* congenital muscular dystrophy alters

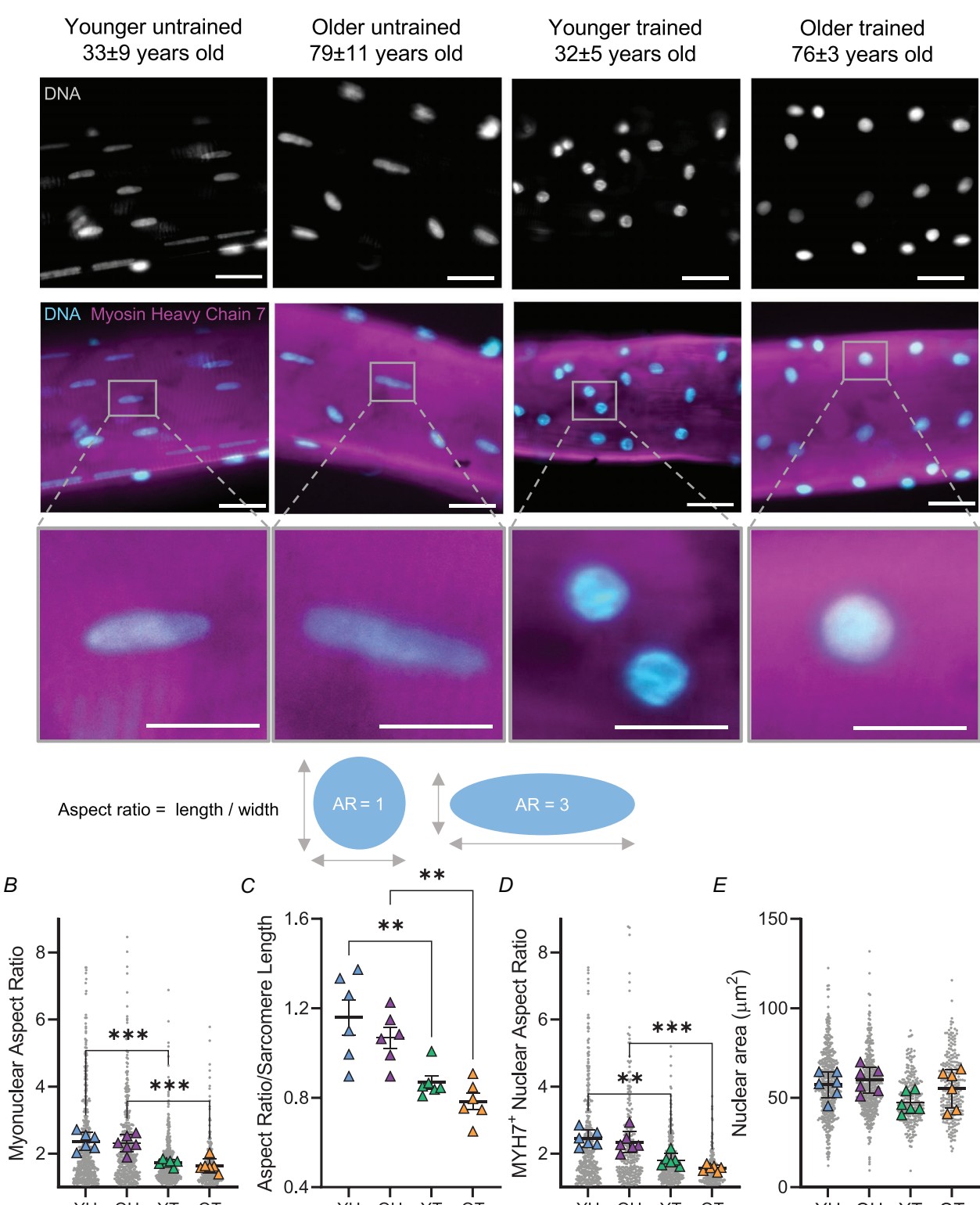

**Figure 1. Altered 2D myonuclear shape in trained younger and older individuals**
*A*, representative images of vastus lateralis muscle fibres isolated from younger untrained (YU), older untrained (OU), younger trained (YT) and older trained (OT) individuals, stained with DAPI and myosin heavy chain 7 to visualise myonuclei and slow myosin, respectively. Scale bars: 30 $\mu$m and 10 $\mu$m main images and zoomed insets of myonuclei, respectively. *B–E*, above graphs, calculation of aspect ratio (length/width of nucleus) and nuclear area ($\mu$m$^2$). *B*, comparisons of myonuclear aspect ratio in YU, OU, YT and OT individuals; 2053 total nuclei analysed. *C*, comparisons of myonuclear aspect ratio between groups after normalisation to sarcomere length. *D*, myonuclear

aspect ratio in MYH7$^+$ fibres (1385 total nuclei analysed). *E*, comparisons of nuclear area ($\mu$m$^2$) between groups (1453 total nuclei analysed). In *B–E*, the coloured symbols represent the mean values for each individual, calculated from several pooled muscle fibres, unfilled grey symbols represent myonuclei; mean values for individuals were used for two-way ANOVA (*n* = 6, \*\**P* < 0.01 \*\*\**P* < 0.001); error bars represent means ± SD. [Colour figure can be viewed at wileyonlinelibrary.com]

mechanotransduction in cultured myotubes and attenuates the hypertrophic response to functional overload in mouse skeletal muscle *in vivo*, implicating a role of lamin A/C in exercise adaptations (Owens et al., 2021). Thus, to investigate whether exercise affects the organisation of lamin A in skeletal muscle, muscle fibres from YU, YT, OU and OT individuals were stained with a lamin A-specific antibody (Fig. 3). As expected, lamin A localised to the periphery of myonuclei (Fig. 3*A*). Importantly, we observed a significant increase in nuclear lamina deposition in myonuclei from OT compared to OU (Fig. 3*B* and *C*). Next, we quantified nuclear invaginations, which are tube-like infoldings of the nuclear envelope and are reported to play roles in premature ageing syndromes (Frost, 2016; McClintock et al., 2006; Schoen et al., 2017). However, our data showed there were no significant differences in total invagination length ($\mu$m) between myonuclei from all groups (Fig. 3*D* and *E*).

These data suggest that training is associated with increased nuclear lamina deposition in primary human skeletal muscle.

### Lamin A levels are increased in exercise-trained mice

Given the precious nature and paucity of human muscle biopsy samples for protein and biophysical analysis, next we used a mouse model to investigate the effects of exercise on myonuclear parameters further.

To determine whether exercise affected the protein levels of nuclear lamins and LINC complex proteins, we performed western blotting on tibialis anterior muscle tissue from mice following 8 weeks of treadmill running. In line with increased lamin A deposition in trained human muscle fibres, lamin A levels were significantly increased in trained mice compared to untrained counterparts (442 ± 53 *vs.* 384 ± 33, respectively, *P* < 0.05; Fig. 4*A*). In contrast, levels of lamins C, B1 and emerin were not significantly different (Fig. 4*A*). LINC complex proteins, which connect the cytoskeleton to the nucleus via the nuclear lamina, were then analysed (Fig. 4*B*). Consistent with the increase in lamin A, levels of Sad1/UNC-84 (SUN) protein 2, SUN2, which is known to preferentially bind lamin A over lamin C (Liang et al., 2011), showed a 45% increase in trained mice that was trending towards significance (1129 ± 498 *vs.* 1639 ± 394, respectively, *P* = 0.055). However, other LINC complex proteins, SUN1, nesprin-1$\alpha$2 and nesprin-2$\alpha$1, were not significantly different in trained compared to untrained mice (Fig. 4*B*).

### Exercise alters myonuclear deformability and stiffness

Changes in nuclear shape and lamin A expression are associated with altered nuclear mechanosensitivity, with lamin A being a key regulator of the mechanical stiffness of nuclei (Lammerding et al., 2006). Thus, we next addressed whether structural alterations in myonuclei from trained individuals translated to biophysical changes in nuclear mechanics. Whilst it is accepted that adaptations to exercise are at least in part driven by mechanotransduction (Attwaters & Hughes, 2022; Hornberger & Esser, 2004; Kirby, 2019), the role of nuclear mechanics in this context has not previously been investigated.

To assess the effects of exercise on myonuclear function, myonuclear deformability was compared in OU and OT human samples. Single muscle fibres were mounted and stretched to different tensions, fixed, and stained to visualise myonuclei and Z-discs (Fig. 5*A*) (adapted from Zhang et al., 2010; Chapman et al., 2014). We reasoned that if nuclear aspect ratio increased proportionately with sarcomere length, nuclei could be considered compliant with fibre tension; conversely, if nuclear aspect ratio did not scale with sarcomere length, they could be considered stiffer. As expected, in both OT and OU fibres, there was a positive relationship between sarcomere length and nuclear aspect ratio (*R*-values were 0.36 and 0.45 for OT and OU, respectively; 211 OT and 260 OU nuclei analysed, respectively; *P* < 0.05; Fig. 5*B*). However, this relationship was significantly steeper in fibres from OU compared to OT (gradients were 0.79 and 0.29, respectively, *P* < 0.05; Fig. 5*B*). Additionally, the variance of myonuclear aspect ratio normalised to sarcomere length was significantly higher in OU compared to OT fibres, showing less consistency in myonuclear shape changes with stretching in OU fibres (Fig. 5*C*, *P* < 0.05).

These data suggest that nuclei in trained individuals were less compliant with increasing fibre tension compared to OU fibres. In other words, nuclei in fibres from trained individuals appeared stiffer than in untrained individuals. To confirm this, we used a nanoindenter (which precisely measures mechanical properties of small samples and cells) to physically probe nuclei and directly test the effects of exercise on myonuclear mechanics (Fig. 5*D* and *E*). Indeed, we observed an 87% increase in Young's modulus (kPa) (a measure of stiffness) in myonuclei from trained mice compared to untrained mice (trained 1.7 ± 0.9 *vs.* untrained 3.2 ± 1.3, *P* < 0.05; Fig. 5*F*). Additionally,

there was a significant difference in the viscoelasticity, whereby tan $\delta$, the fraction of the loss modulus (the viscous component) over the storage modulus (the elastic component), was on average ~20% lower in exercise trained mice at 1, 2 and 4 Hz DMA ($P < 0.05$), and 14% lower at 10 Hz DMA (not significant) (Fig. 5*G–I*), indicating more elastic nuclei in the trained mice.

Taken together, analysis of nuclear mechanics in human and mouse muscle fibres indicated that training

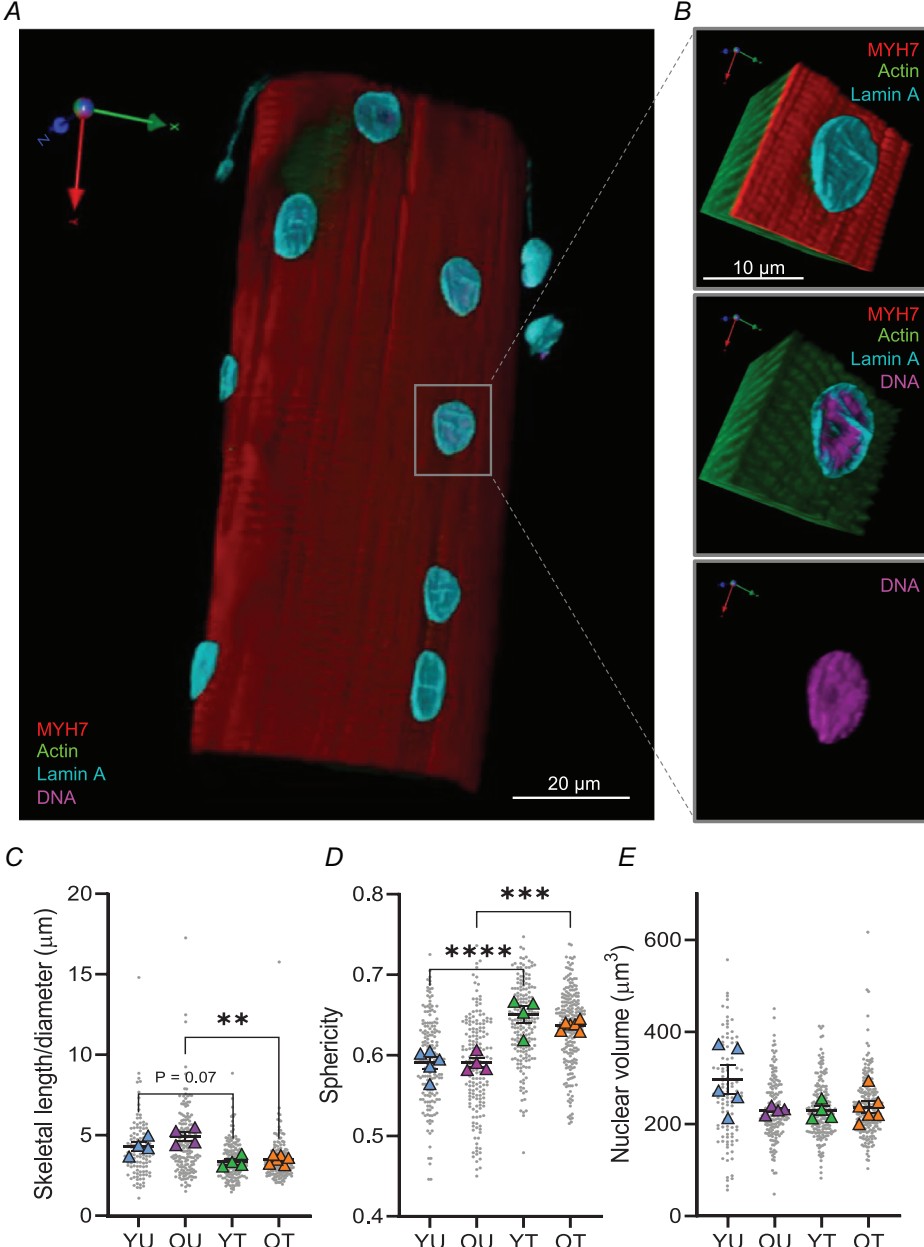

**Figure 2. Lower 3D aspect ratio and greater sphericity in myonuclei from trained younger and older individuals**

*A*, representative three-dimensional rendering of Z-stack images of a single human vastus lateralis muscle fibre acquired with a spinning disk confocal microscope equipped with a ×63 oil objective lens. Muscle fibre stained to visualise lamin A, DNA, actin and myosin heavy chain 7 (MYH7). *B*, representative zoomed images of 3D-rendered nucleus. *C–E*, comparisons of nuclear skeletal length/diameter ($\mu$m), sphericity and volume in younger untrained (YU), older untrained (OU), younger trained (YT) and older trained (OT) individuals. Coloured symbols represent individual means, grey symbols represent myonuclei; mean values for individuals were used for two-way ANOVA ($n = 4$ to 6, **$P < 0.01$ ***$P < 0.001$ ****$P < 0.0001$); error bars represent means $\pm$ SD. [Colour figure can be viewed at wileyonlinelibrary.com]

reduced myonuclear deformability and increased myonuclear stiffness and elasticity.

## Discussion

Muscle pathologies and premature ageing syndromes caused by mutations in nuclear lamina and envelope proteins have revealed a common phenotype: abnormal nuclear shape and defective mechanotransduction. However, whether similar structural and functional defects occur with physiological ageing with and without exercise in human skeletal muscle had not previously been investigated.

Our main findings are that exercise, regardless of age, is associated with more spherical, less deformable myonuclei, with increased lamin A levels and deposition at the nuclear lamina. This implies that myonuclear mechanotransduction may have a role in governing exercise adaptations (Fig. 6A). Maintaining myonuclear structure and function through regular exercise may be an important factor in preserving muscle function

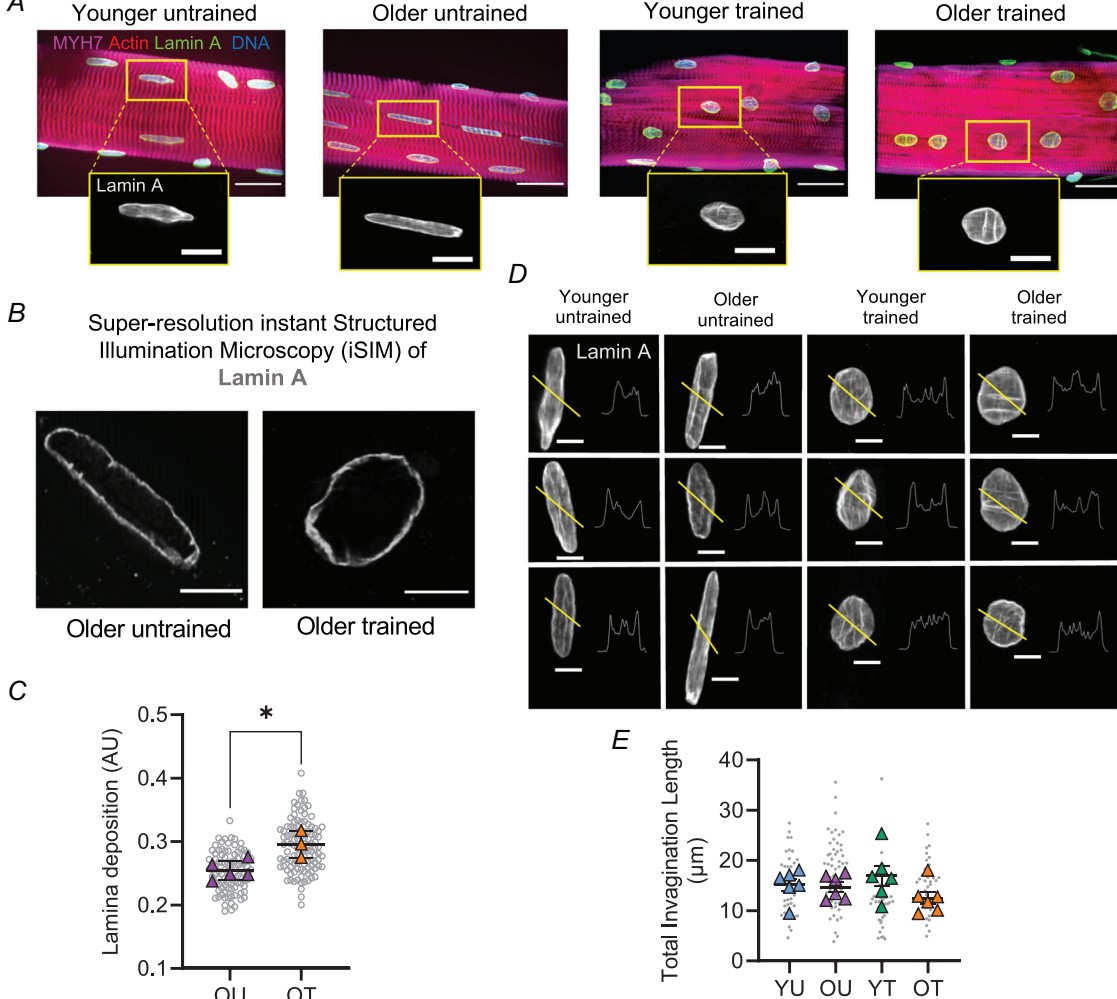

**Figure 3. Organisation of lamin A in trained individuals and untrained counterparts**
*A*, representative images, acquired through confocal microscopy using a ×63 oil objective, of muscle fibres isolated from younger untrained (YU), older untrained (OU) patients, younger trained (YT) and older untrained (OT) individuals. Fibres were stained with DAPI to visualise DNA, actin, lamin A and myosin heavy chain 7 (MYH7). Scale bar: 25 $\mu$m in main images, 10 $\mu$m in zoomed images. *B*, representative images of myonuclei from OU and OT muscle fibres acquired through super-resolution iSIM microscopy. Scale bars: 5 $\mu$m. *C*, quantification of lamin A deposition ($\mu$m) in muscle fibres from OT and OU. *n* = 3–5 per group; unpaired *t*-test revealed significant difference between groups (P < 0.05). *D*, standard deviation projections of lamin A-stained myonuclei and pixel intensity line scans from YU, OU, YT and OT. *E*, lamin A total invagination length ($\mu$m) in muscle fibres from YU, OU, YT and OT; *n* = 6. Two-way ANOVA revealed no significant differences between groups. Coloured symbols represent individual means, grey symbols represent myonuclei; mean values for individuals were used for two-way ANOVA and *t*-test. Error bars represent means ± SD. [Colour figure can be viewed at wileyonlinelibrary.com]

throughout the lifespan (Fig. 6*B*). Conversely, myonuclear dysfunction in untrained muscle may contribute to age-related decline in muscle mass and function (Fig. 6*B*). Below, possible mechanisms and consequences of exercise-related and inactivity-related alterations in myonuclear structure and mechanics will be discussed.

Greater myonuclear sphericity and stiffness in trained muscle fibres may be a result of the increased lamin A expression. Lamin A regulates myonuclear shape and mechanics in muscle cells, with loss or misexpression of lamin A resulting in myonuclear elongation and increased deformability (Earle et al., 2020; Roman et al., 2017). Forces can also induce conformational changes to nuclear envelope and lamina proteins, modulating the mechanical properties of nuclei (Buxboim et al., 2014; Guilluy et al., 2014; Swift et al., 2013). Thus, increased

lamin A expression may be an initial response to exercise training, which causes increased myonuclear sphericity and stiffness, and reduces myonuclear deformability in trained muscle fibres.

The structural and mechanical alterations to myonuclei in trained individuals may have several consequences beneficial to muscle fibre function (Fig. 6). These alterations may be underpinned by altered chromatin organisation and expression of genes important for oxidative capacity or repression of atrophy (Fischer et al., 2016; Ho et al., 2013). More spherical, stiffer myonuclei with increased lamin A expression in trained individuals may have enhanced transduction of forces via the nuclear lamina, which tethers chromatin (Schreiner et al., 2015). This may directly regulate the expression of genes important for exercise adaptations, by stretching

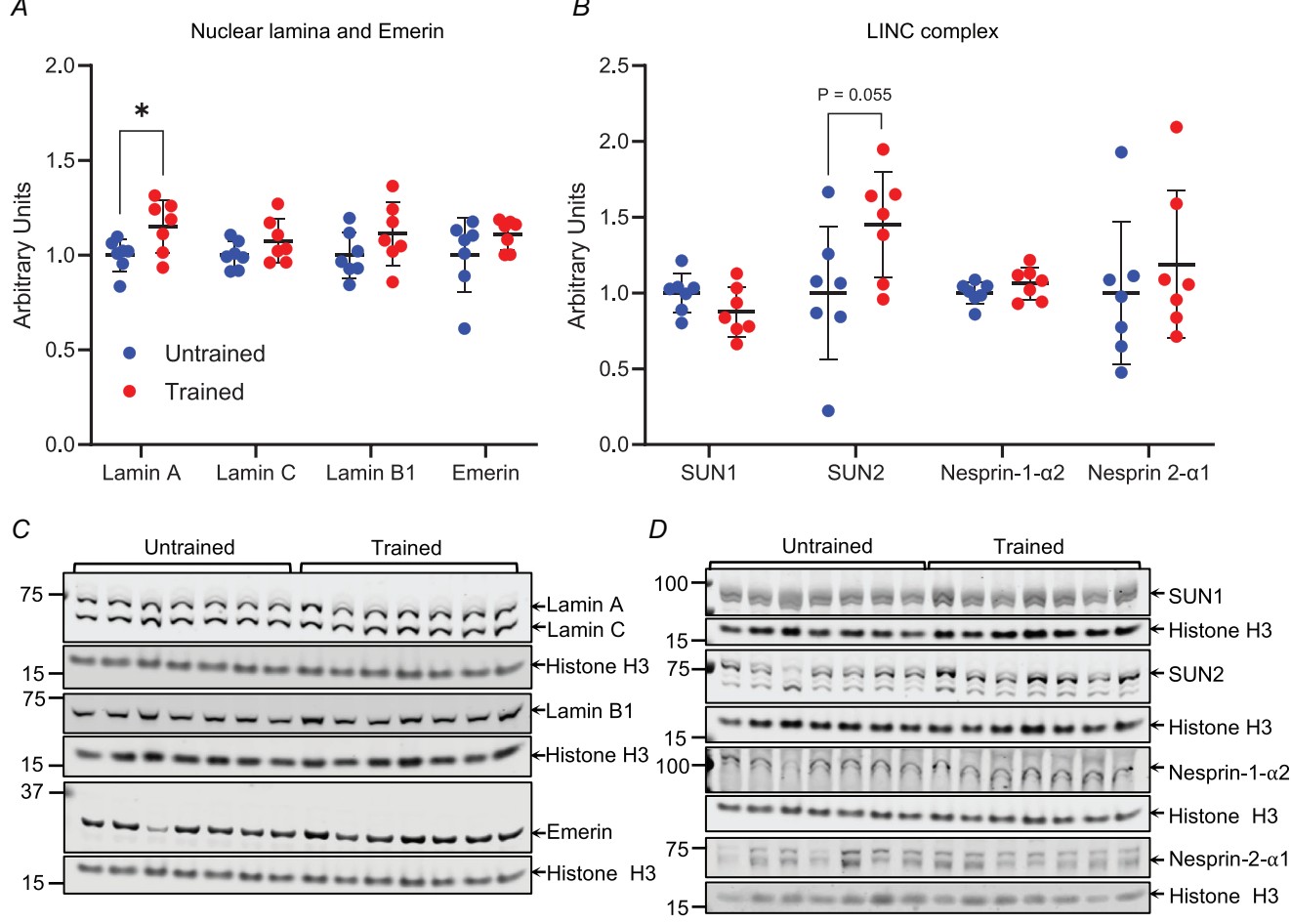

**Figure 4. Lamin A levels are increased in trained mouse tibialis anterior muscle**
*A*, protein levels of lamin A, lamin C, lamin B1 and emerin normalised to histone H3 in tibialis anterior muscle from untrained and high intensity endurance trained mice. *B*, protein levels of linker of nucleoskeleton and cytoskeleton (LINC) complex proteins SUN1, SUN2, nesprin-1-α2 and nesprin-2-α1 normalised to histone H3 in tibialis anterior muscle from untrained and trained mice. *C* and *D*, images of western blots from which data in *A* and *B* were obtained. Note that lamin A levels were significantly increased and SUN2 levels trending to increase. Arrows indicate predicted molecular masses (kDa). Data points represent individual mice; *n* = 7 per group; **P* < 0.05 (*t*-test). Error bars represent means ± SD. [Colour figure can be viewed at wileyonlinelibrary.com]

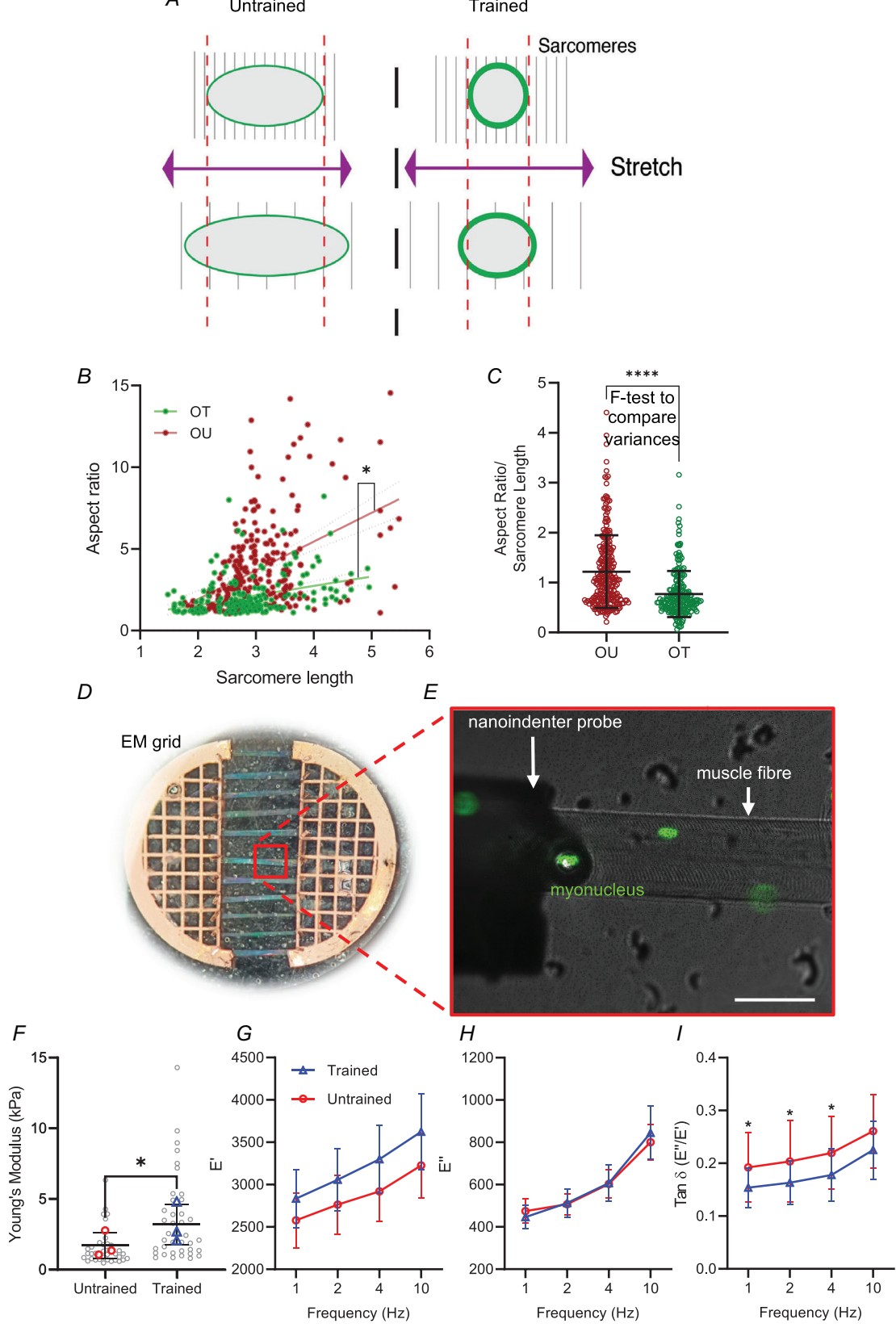

**Figure 5. Exercise training results in stiffer myonuclei**

*A*, schematic representation of fibre mounting and stretching. *B*, relationship between sarcomere length and nuclear aspect ratio in muscle fibres from older trained (OT) and untrained (OU) patients. *C*, variance of aspect

ratio/sarcomere length in OT and OU fibres. Note that sarcomere length positively correlates with extent of muscle fibre stretch and that myonuclei in OT fibres were significantly stiffer than myonuclei from OU fibres. *Statistically significant difference between gradients of slopes (linear regression analysis, *F*-test to compare variances). *D*, typical set-up of individual muscle fibres after isolation and mounting in parallel on an electron microscopy (EM) grid split in half for imaging and nanoindentation. *E*, nanoindentation of single muscle fibres (brightfield) with myonuclei labelled being probed by nanoindenter (brightfield, left of image). Scale bar, 50 $\mu$m. *F*, comparison of Young's modulus (kPa) in nuclei from untrained and trained mice. *G–I*, comparisons of *E′*, *E″* and tan $\delta$ (*E′/E′*) at different dynamic mechanical analysis frequencies (Hz) in nuclei from untrained and exercise trained mice. Note that myonuclei were significantly more stiff and more elastic in fibres from trained *vs.* untrained mice. Each coloured data point represents the average for each mouse, *n* = 3 per group. Error bars represent means ± SEM. *Statistically significant difference between groups (*t*-test and mixed effects analysis). [Colour figure can be viewed at wileyonlinelibrary.com]

or compacting chromatin to alter transcription factor accessibility (Tajik et al., 2016). Additionally, increased nuclear stiffness and altered lamina composition may affect translocation of transcription factors into the nucleus, given the responsiveness of nuclear pore complexes to cellular forces (Sapra et al., 2020; Zimmerli et al., 2021). For example, greater lamin A expression may increase translocation of transcriptional co-factors Yap/Taz, which are associated with muscle growth and signalling pathways involved in both endurance and resistance exercise adaptations (Gabriel et al., 2016). Furthermore, altered lamin A levels may affect nucleo-cytoskeletal shuttling of mechanosensitive transcription factor MRTF-A as reported in lamin A/C-null mouse embryonic fibroblasts (Ho et al., 2013).

In addition to facilitating exercise adaptations, myonuclear remodelling in trained individuals may be mechanoprotective. Conversely, nuclear defects driven by inactivity may be detrimental for cellular function and health (Kalukula et al., 2022). Thus, increased lamin A expression and stiffness, and reduced deformability of myonuclei in trained muscle fibres may improve resilience against contractile forces during future exercise bouts.

The elongated shape and greater deformability of myonuclei in untrained individuals were reminiscent of those in muscle fibres from humans and mice with muscular dystrophies characterised by muscle wasting and dysfunction (Earle et al., 2020; Tan et al., 2015). Thus, defective myonuclear structure and function due to inactivity may contribute to age-related muscle dysfunction. Specifically, chromatin stretching in more compliant myonuclei may result in expression of genes that contribute to muscle atrophy, which are elevated after 2 weeks of inactivity (Jones et al., 2004). Additionally, altered chromatin organisation may repress genes encoding contractile or mitochondrial proteins, decreasing force production and endurance capacity (Fig. 6*B*). Deformable myonuclei in untrained individuals may be more susceptible to nuclear envelope rupture, impacting cell health (Earle et al., 2020; Kalukula et al., 2022). These possible consequences of myonuclear dysfunction in old age may collectively contribute to impairments in muscle mass, strength and endurance

with age, and be alleviated by exercise-mediated myonuclear remodelling (Fig. 6*B*).

A limitation of our work was the limited availability of human muscle biopsies from the four groups, which could be complemented with future studies. In the present investigation, the older untrained group was composed of hip fracture patients with other underlying health conditions (see Table 1) which may have influenced the observed myonuclear aberrations. Participants were mixed sex, and drug administration and variations in habitual dietary intake were not stringently accounted for, possibly introducing variability in muscle fibre size and other variables. To this end, a study group composed of an older untrained group with clearer inclusion and exclusion criteria related to physical activity levels may provide a more accurate representation of the consequences of inactive ageing. Nevertheless, analysis of muscle fibres from this group has provided insight into myonuclear structure and function in elderly inactive individuals. Furthermore, these individuals displayed a strikingly similar nuclear shape phenotype to apparently healthy, younger untrained individuals.

The laborious and time-consuming nature of recruiting human participants and acquiring their samples for longitudinal studies makes cross-sectional analyses more feasible. To confirm our findings, a longitudinal study of myonuclear structure and function in samples serially acquired from active and inactive individuals throughout their lifespan would be required. This would provide a comprehensive account of the effects of active and inactive ageing on skeletal muscle myonuclei, and a causal relationship between changes in myonuclear parameters and muscle function.

In summary, our data suggest that exercise is associated with profound alterations in nuclear structure and mechanics in human primary muscle fibres regardless of age. In line with this, exercise resulted in increased lamin A expression and myonuclear stiffness in mice. Future investigations into the potential role of myonuclear mechanotransduction in exercise and ageing would further our understanding of skeletal muscle physiology and offer new insights into improving human healthspan.

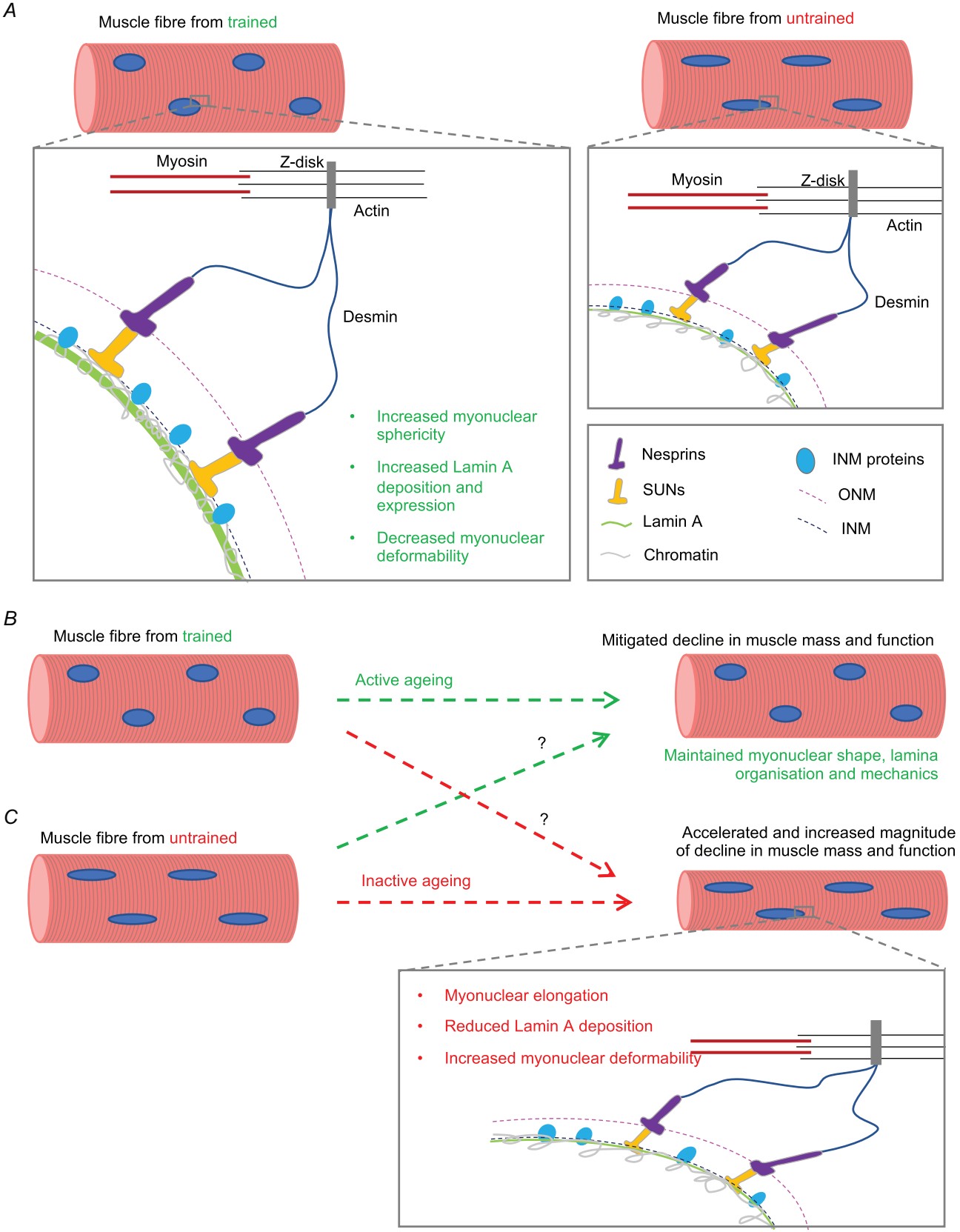

**Figure 6. Summary and proposed effects of myonuclear remodelling with training, and inactivity related defects in nuclear mechanotransduction with age**

*A*, in skeletal muscle fibres from trained individuals, nuclear envelope proteins including the nuclear lamina effectively transduce cytoskeletal forces to the nucleus to regulate signalling pathways. *B*, in skeletal muscle from trained older individuals, myonuclear shape and mechanotransduction are preserved. *C*, in skeletal muscle from untrained older individuals, myonuclei are more elongated, nuclear lamina levels are reduced, and myonuclei are more deformable. This may lead to increased susceptibility to myonuclear damage and defective mechanotransduction that results in decline in muscle mass and function. [Colour figure can be viewed at wileyonlinelibrary.com]

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

## Additional information

### Data availability statement

Individual datapoints ($n \leq 30$) are included in the figures. Data from the study will be made available upon reasonable request.

## Competing interests

None.

## Author contributions

M.J.S. and J.O. contributed to the conception of the work. E.B., J.A.R., M.J.S. and J.O. designed experiments. E.B., J.A.R., A.H., Y.H. and D.G.S.W. performed experiments. E.B., A.H., D.G.S.W. and T.I. completed the formal analysis of the data. E.B., J.A.R., A.H., D.G.S.W., T.I., J.O. and M.J.S. interpreted the data. E.B. completed data visualisation. R.D.P., M.K., J.N.P., G.L.C., N.R.L., S.D.R.H. and J.O. recruited human participants and collected human muscle biopsy samples. E.B. and M.J.S. wrote the first draft of the manuscript. E.B., J.A.R., A.H., R.D.P., M.K., G.M.E., N.R.L., T.I., S.D.R.H., J.O. and M.J.S. contributed to the manuscript and revised it critically. All authors have read and approved the final version of this manuscript and agree to be accountable for all aspects of the work in ensuring that questions related to the accuracy or integrity of any part of the work are appropriately investigated and resolved. All persons designated as authors qualify for authorship, and all those who qualify for authorship are listed.

## Funding

J.O. and E.B. are funded by the Medical Research Council of the UK (MR/S023593/1). M.J.S. is supported by British Heart Foundation Intermediate Fellowship: FS/17/57/32934 and King's BHF Centre for Excellence Award: RE/18/2/34213. S.D.R.H., N.R.L. and R.D.P. were funded by the Bupa Foundation. M.K. was funded by a King's College London PhD studentship.

## Keywords

ageing, exercise, nuclear lamina, nuclear shape, nuclei

## Supporting information

Additional supporting information can be found online in the Supporting Information section at the end of the HTML view of the article. Supporting information files available:

**Statistical Summary Document**
**Peer Review History**

