## [Peer Review History · The Journal of Physiology]

Myonuclear alterations associated with exercise are independent of age in humans

Edmund Battey, Jacob A Ross, Anthony Hoang, Darren G S Wilson, Yu Han, Yotam Levy, Ross D Pollock, Michaeljohn Kalakoutis, Jamie Pugh, Graeme L Close, Georgina M Ellison-Hughes, Norman R Lazarus, Thomas Iskratsch, Stephen D. R. Harridge, Julien Ochala, and Matthew J Stroud

DOI: 10.1113/JP284128

Corresponding author(s): Matthew Stroud (matthew.stroud@kcl.ac.uk)

The following individual(s) involved in review of this submission have agreed to reveal their identity: Cory Dungan (Referee #1); William Roman (Referee #2)

Review Timeline:

Submission Date:	17-Nov-2022
Editorial Decision:	29-Nov-2022
Revision Received:	06-Dec-2022
Accepted:	19-Dec-2022

Senior Editor: Michael Hogan

Reviewing Editor: Kevin Murach

Transaction Report:

Dear Dr Stroud,

Re: JP-RP-2022-284128 "Myonuclear alterations associated with exercise are independent of age in humans" by Edmund Battey, Jacob A Ross, Anthony Hoang, Darren G S Wilson, Yu Han, Yotam Levy, Ross D Pollock, Michaeljohn Kalakoutis, Jamie Pugh, Graeme L Close, Georgina M Ellison, Norman R Lazarus, Thomas Iskratsch, Stephen D. R. Harridge, Julien Ochala, and Matthew J Stroud

Thank you for submitting your manuscript to The Journal of Physiology. It has been assessed by a Reviewing Editor and by 2 expert referees and we are pleased to tell you that it is acceptable for publication following minor revision.

REVISION CHECKLIST:

We look forward to receiving your revised submission.

Yours sincerely,

Michael C. Hogan
Senior Editor
The Journal of Physiology
<https://jp.msubmit.net>
<http://jp.physoc.org>
The Physiological Society
Hodgkin Huxley House
30 Farringdon Lane
London, EC1R 3AW
UK
<http://www.physoc.org>
<http://journals.physoc.org>

REQUIRED ITEMS:

-Author photo and profile. First (or joint first) authors are asked to provide a short biography (no more than 100 words for one author or 150 words in total for joint first authors) and a portrait photograph. These should be uploaded and clearly labelled with the revised version of the manuscript. See Information for Authors for further details.

-You must start the Methods section with a paragraph headed Ethical Approval. A detailed explanation of journal policy and regulations on animal experimentation is given in Principles and standards for reporting animal experiments in The Journal of Physiology and Experimental Physiology by David Grundy J Physiol, 593: 2547-2549. doi:10.1113/JP270818.). A checklist outlining these requirements and detailing the information that must be provided in the paper can be found at: <https://physoc.onlinelibrary.wiley.com/hub/animal-experiments>. Authors should confirm in their Methods section that their experiments were carried out according to the guidelines laid down by their institution's animal welfare committee, and conform to the principles and regulations as described in the Editorial by Grundy (2015). The Methods section must contain details of the anaesthetic regime: anaesthetic used, dose and route of administration and method of killing the experimental animals.

-You must start the Methods section with a paragraph headed Ethical Approval. If experiments were conducted on humans confirmation that informed consent was obtained, preferably in writing, that the studies conformed to the standards set by the latest revision of the Declaration of Helsinki, and that the procedures were approved by a properly constituted ethics committee, which should be named, must be included in the article file. If the research study was registered (clause 35 of the Declaration of Helsinki) the registration database should be indicated, otherwise the lack of registration should be noted as an exception (e.g. The study conformed to the standards set by the Declaration of Helsinki, except for registration in a database.). For further information see: <https://physoc.onlinelibrary.wiley.com/hub/human-experiments>

-Your manuscript must include a complete Additional Information section

-Please upload separate high-quality figure files via the submission form.

-You must upload original, uncropped western blot/gel images (including controls) if they are not included in the manuscript. This is to confirm that no inappropriate, unethical or misleading image manipulation has occurred <https://physoc.onlinelibrary.wiley.com/hub/journal-policies#imagmanip> These should be uploaded as 'Supporting information

for review process only'. Please label/highlight the original gels so that we can clearly see which sections/lanes have been used in the manuscript figures.

-Please ensure that any tables are in Word format and are, wherever possible, embedded in the article file itself.

-Please ensure that the Article File you upload is a Word file.

-Your paper contains Supporting Information of a type that we no longer publish. Any information essential to an understanding of the paper must be included as part of the main manuscript and figures. The only Supporting Information that we publish are video and audio, 3D structures, program codes and large data files. Your revised paper will be returned to you if it does not adhere to our Supporting Information Guidelines

-A Statistical Summary Document, summarising the statistics presented in the manuscript, is required upon revision. It must be on the Journal's template, which can be downloaded from the link in the Statistical Summary Document section here: https://jp.msubmit.net/cgi-bin/main.plex?form_type=display_requirements#statistics

-Papers must comply with the Statistics Policy https://jp.msubmit.net/cgi-bin/main.plex?form_type=display_requirements#statistics

In summary:

-If $n \leq 30$, all data points must be plotted in the figure in a way that reveals their range and distribution. A bar graph with data points overlaid, a box and whisker plot or a violin plot (preferably with data points included) are acceptable formats.

-If $n > 30$, then the entire raw dataset must be made available either as supporting information, or hosted on a not-for-profit repository e.g. FigShare, with access details provided in the manuscript.

- n clearly defined (e.g. x cells from y slices in z animals) in the Methods. Authors should be mindful of pseudoreplication.

-All relevant n values must be clearly stated in the main text, figures and tables, and the Statistical Summary Document (required upon revision)

-The most appropriate summary statistic (e.g. mean or median and standard deviation) must be used. Standard Error of the Mean (SEM) alone is not permitted.

-Exact p values must be stated. Authors must not use 'greater than' or 'less than'. Exact p values must be stated to three significant figures even when 'no statistical significance' is claimed.

-Statistics Summary Document completed appropriately upon revision

-Please include an Abstract Figure file, as well as the figure legend text within the main article file. The Abstract Figure is a piece of artwork designed to give readers an immediate understanding of the research and should summarise the main conclusions. If possible, the image should be easily 'readable' from left to right or top to bottom. It should show the physiological relevance of the manuscript so readers can assess the importance and content of its findings. Abstract Figures should not merely recapitulate other figures in the manuscript. Please try to keep the diagram as simple as possible and without superfluous information that may distract from the main conclusion(s). Abstract Figures must be provided by authors no later than the revised manuscript stage and should be uploaded as a separate file during online submission labelled as File Type 'Abstract Figure'. Please ensure that you include the figure legend in the main article file. All Abstract Figures should be created using BioRender. Authors should use The Journal's premium BioRender account to export high-resolution images. Details on how to use and access the premium account are included as part of this email.

EDITOR COMMENTS

Reviewing Editor:

Your work has been evaluated by two experts in the field. Both reviewers found your work to be novel and interesting. One reviewer requested some additional experimentation. While I agree that additional experimentation could provide important information on the consequences of altering nuclear morphology, such experiments would significantly extend the time to publication and could be saved for follow-up investigations. Please respond to the reviewer's consequences as comprehensively as possible.

REFeree COMMENTS

Referee #1:

Bathey and colleagues examine myonuclear shape and mechanics in skeletal muscle from young and old humans following exercise and provide novel insight on how age and exercise affect skeletal muscle. This work is innovative and physiologically relevant, and provides new insight into the physiologic adaptations to exercise.

- First, I would like to commend the authors for their detailed analysis and thorough methods.
- The authors mention that the adaptations to exercise are, in part, mediated by mechanotransduction and provide a brief discussion on how this could affect gene expression. Due to changes in shape and stiffness being central to this manuscript, I am hoping that the authors can expand upon this section in the discussion and provide a little more detail on how changes in stiffness affects mechanotransduction-mediated gene expression.
- It would be interesting to hear the authors' thoughts on how changes in stiffness and myonuclear size with aging could contribute to anabolic resistance during aging.
- For some of the outcomes (i.e., Figure 1C, Figure 5C) the authors normalized data relative to the sarcomere length. Although the methods are quite detailed, which is appreciated by this reviewer, I am not entirely certain how sarcomere length was quantified (method, software used, number counted per muscle fiber, etc.). It would be good to have a little more detail on this in the methods.
- The authors should add detail on their animal procedures (i.e., source of animals, access to food and water, temperature, etc.).
- A recent manuscript by Rader et al. (<https://physoc.onlinelibrary.wiley.com/doi/full/10.14814/phy2.15476>) appears relevant to this work and should be added to the citations.

Referee #2:

In this manuscript, Bathey et al. assess the morphological and functional differences between myonuclei from trained and untrained muscle in young and aged individuals. The first portion of the manuscript shows that untrained individuals possess less round/spherical myonuclei when compared to muscle obtained from trained patients. These changes are consistent in both young and aged muscle. By staining isolated fibers for lamin A, the authors also observe a decrease in lamin A in old untrained individuals when compared to their trained counterparts. They deepen this analysis of LINC complex protein alterations by showing increased expression of lamin A and SUN2 in trained mice. Finally, they perform functional assays to show that trained myonuclei from aged individuals are stiffer and less deformable.

The conclusions of the study are supported by the evidence provided. The manuscript follows a clear and concise demonstration, and the text is well written. The data displayed is of high quality and the efforts put to rely on human samples for the majority of the work is impressive and will be appreciated by the community. The observations made could be influential in the fields of exercise physiology and muscle disorders associated with nuclear morphology defects. This does

leave the readers awaiting a bit more such as an assessment of DNA damage or nuclear envelope rupture, alterations in gene expression or mechanotransduction signalling. The authors do discuss some of these potential consequences, but experimental explorations would substantially heighten the scope of the manuscript. Overall, this study is well performed and relevant to the muscle field.

Major points:

As eluded to in the general comments, the authors could strengthen their work by demonstrating the consequences of altered nuclear morphology and lower lamin A levels in untrained myonuclei (DNA damage, nuclear membrane instability, gene expression, etc...). The authors do highlight limitations in human sample quantity thereby hampering a timely investigation of these morphological changes. Nevertheless, this could be performed in mouse as trained mice display increased lamin A and SUN2 expression as well as higher nuclear stiffness. It is unclear if the morphology of mice myonuclei is similarly altered than what is observed in humans but a few key experiments could be done in mice to show potential effects. For example if one session of exercise leads to higher nuclear damage in untrained animals.

Minor points:

Figure 1:

- The plots are very informative and aesthetic. Could you detail if the colored symbols represent individual means from a single myofiber or from several myofibers pooled in the same experiment.
- Panel C shows the normalization of myonuclear aspect ratio over sarcomere length to show no change in trend when taking that factor into account. Can the authors comment if the sarcomere length alone is different in the various cohorts?

Figure 2:

- The 3D aspect ratio is not as efficiently defined in 3D as it is in 2D. Can the authors better describe the 3D aspect ratio?
- Can the authors provide a 3D representative image (similar to Fig2.A) of trained and untrained isolated fibers for comparison.
- The panels are mis referenced in the text which has made the understanding of this figure more difficult.

Figure 3:

- The sentence "Thus, to investigate whether exercise affects the organisation of lamin A..." seems grammatically off. Adding line numbers would also help the revision process.

Figure 5:

- A quick description of a nanoindenter within the text would help the reader understand the assay.

END OF COMMENTS

Confidential Review

17-Nov-2022

Batley and colleagues examine myonuclear shape and mechanics in skeletal muscle from young and old humans following exercise and provide novel insight on how age and exercise affect skeletal muscle. This work is innovative and physiologically relevant, and provides new insight into the physiologic adaptations to exercise.

- First, I would like to commend the authors for their detailed analysis and thorough methods.
- The authors mention that the adaptations to exercise are, in part, mediated by mechanotransduction and provide a brief discussion onto how this could affect gene expression. Due to changes in shape and stiffness being central to this manuscript, I am hoping that the authors can expand upon this section in the discussion and provide a little more detail on how changes in stiffness affects mechanotransduction-mediated gene expression.
- It would be interesting to hear the authors thoughts on how changes in stiffness and myonuclear size with aging could contribute to anabolic resistance during aging.
- For some of the outcomes (i.e., Figure 1C, Figure 5C) the authors normalized data relative to the sarcomere length. Although the methods are quite detailed, which is appreciated by this reviewer, I am not entirely certain how sarcomere length was quantified (method, software used, number counted per muscle fiber, etc.). It would be good to have a little more detail on this in the methods.
- The authors should add detail on their animal procedures (i.e., source of animals, access to food and water, temperature, etc.).
- A recent manuscript by Rader et al. (<https://physoc.onlinelibrary.wiley.com/doi/full/10.14814/phy2.15476>) appears relevant to this work and should be added to the citations.

In this manuscript, Battey et al. assess the morphological and functional differences between myonuclei from trained and untrained muscle in young and aged individuals. The first portion of the manuscript shows that untrained individuals possess less round/spherical myonuclei when compared to muscle obtained from trained patients. These changes are consistent in both young and aged muscle. By staining isolated fibers for lamin A, the authors also observe a decrease in lamin A in old untrained individuals when compared to their trained counter parts. They deepen this analysis of LINC complex proteins alterations by showing increased expression of lamin A and SUN2 in trained mice. Finally, they perform functional assays to show that trained myonuclei from aged individuals are stiffer and less deformable.

The conclusions of the study are supported by the evidence provided. The manuscript follows a clear and concise demonstration, and the text is well written. The data displayed is of high quality and the efforts put to rely on human samples for the majority of the work is impressive and will be appreciated by the community. The observations made could be influential in the fields of exercise physiology and muscle disorders associated with nuclear morphology defects. This does leave the readers awaiting a bit more such as an assessment of DNA damage or nuclear envelope rupture, alterations in gene expression or mechanotransduction signalling. The authors do discuss some of these potential consequences, but experimental explorations would substantially heighten the scope of the manuscript. Overall, this study is well performed and relevant to the muscle field.

Major points:

As eluded to in the general comments, the authors could strengthen their work by demonstrating the consequences of altered nuclear morphology and lower lamin A levels in untrained myonuclei (DNA damage, nuclear membrane instability, gene expression, etc...). The authors do highlight limitations in human sample quantity thereby hampering a timely investigation of these morphological changes. Nevertheless, this could be performed in mouse as trained mice display increased lamin A and SUN2 expression as well as higher nuclear stiffness. It is unclear if the morphology of mice myonuclei is similarly altered than what is observed in humans but a few key experiments could be done in mice to show potential effects. For example if one session of exercise leads to higher nuclear damage in untrained animals.

Minor points:

Figure1:

- The plots are very informative and aesthetic. Could you detail if the colored symbols represent individual means from a single myofiber or from several myofibers pooled in the same experiment.
- Panel C shows the normalization of myonuclear aspect ratio over sarcomere length to show no change in trend when taking that factor into account. Can the authors comment if the sarcomere length alone is different in the various cohorts?

Figure 2:

- The 3D aspect ratio is not as efficiently defined in 3D as it is in 2D. Can the authors better describe the 3D aspect ratio?

- Can the authors provide a 3D representative image (similar to Fig2.A) of trained and untrained isolated fibers for comparison.
- The panels are mis referenced in the text which has made the understanding of this figure more difficult.

Figure 3:

- The sentence “Thus, to investigate whether exercise affects the organisation of lamin A...” seems grammatically off. Adding line numbers would also help the revision process.

Figure 5:

- A quick description of a nanoindenter within the text would help the reader understand the assay.

Dear Dr Stroud,

Re: JP-RP-2022-284128 "Myonuclear alterations associated with exercise are independent of age in humans" by Edmund Battey, Jacob A Ross, Anthony Hoang, Darren G S Wilson, Yu Han, Yotam Levy, Ross D Pollock, Michaeljohn Kalakoutis, Jamie Pugh, Graeme L Close, Georgina M Ellison, Norman R Lazarus, Thomas Iskratsch, Stephen D. R. Harridge, Julien Ochala, and Matthew J Stroud

Thank you for submitting your manuscript to The Journal of Physiology. It has been assessed by a Reviewing Editor and by 2 expert referees and we are pleased to tell you that it is acceptable for publication following minor revision.

Your revised manuscript should be submitted online using the link in your Author Tasks: <https://jp.msubmit.net/cgi-bin/main.plex?el=A1JS1FUm3A7bgD3F3A9ftdfIDVjn5jSZL68pZ6MMqEGQZ>. This link is accessible via your account as Corresponding Author; it is not available to your co-authors. If this presents a problem, please contact journal staff (jp@physoc.org). Image files from the previous version are retained on the system. Please ensure you replace or remove any files that are being revised.

This will enable Authors to create and download high-resolution figures. If authors have used the free BioRender service, they can use the instructions provided in the link above to

download a high-resolution version suitable for publication.

LANGUAGE EDITING AND SUPPORT FOR PUBLICATION: If you would like help with English language editing, or other article preparation support, Wiley Editing Services offers expert help, including English Language Editing, as well as translation, manuscript formatting, and figure formatting at www.wileyauthors.com/eoo/preparation. You can also find resources for Preparing Your Article for general guidance about writing and preparing your manuscript at www.wileyauthors.com/eoo/prepresources.

REQUIRED ITEMS:

-Author photo and profile. First (or joint first) authors are asked to provide a short biography (no more than 100 words for one author or 150 words in total for joint first authors) and a portrait photograph. These should be uploaded and clearly labelled with the revised version of the manuscript. See Information for Authors for further details.

-You must start the Methods section with a paragraph headed Ethical Approval. A detailed explanation of journal policy and regulations on animal experimentation is given in Principles and standards for reporting animal experiments in The Journal of Physiology and Experimental Physiology by David Grundy J Physiol, 593: 2547-2549. doi:10.1113/JP270818.). A checklist outlining these requirements and detailing the information that must be provided in the paper can be found at: <https://physoc.onlinelibrary.wiley.com/hub/animal-experiments>. Authors should confirm in their Methods section that their experiments were carried out according to the guidelines laid down by their institution's animal welfare committee, and conform to the principles and regulations as described in the Editorial by Grundy (2015). The Methods section must contain details of the anaesthetic regime: anaesthetic used, dose and route of administration and method of killing the experimental animals.

-You must start the Methods section with a paragraph headed Ethical Approval. If experiments were conducted on humans confirmation that informed consent was obtained, preferably in writing, that the studies conformed to the standards set by the latest revision of the Declaration of Helsinki, and that the procedures were approved by a properly constituted ethics committee, which should be named, must be included in the article file. If the research study was registered (clause 35 of the Declaration of Helsinki) the registration database should be indicated, otherwise the lack of registration should be noted as an exception (e.g. The study conformed to the standards set by the Declaration of Helsinki, except for registration in a database.). For further information see: <https://physoc.onlinelibrary.wiley.com/hub/human-experiments>

We have included a paragraph at the start of the methods section headed Ethical approval, and have also included a paragraph headed Animal handling (pasted below).

Ethical approval

Prior to participation, written informed consent was obtained from all subjects. Procedures were approved by the Fulham Research Ethics Committee in London (12/LO/0457), Westminster Ethics Committee in London (12/LO/0457) or Liverpool John Moores ethics committee (H17SPS012) and conformed to the standards set by the Declaration of Helsinki. All human tissues were collected, stored, and analysed in accordance with the Human Tissue Act. Procedures were performed in accordance with the Guidance on the Operation of the Animals (Scientific Procedures) Act, 1986 (UK Home Office); King's College London License number: X24D82DFF, ethics code: PDB33C80B.

Animal handling

24 male C57BL/6 mice were maintained in groups of four in cages, lined with wood shavings, cardboard rolls and cleaned weekly, in an automated room for photoperiod control (light-dark cycle 12 h/ 12 h). Animals were provided with water and a standard chow diet ad libitum. 12 mice were trained on treadmill over 8 weeks, with a complementary sedentary group left in cages for an equivalent time-period. At the end of the 8 weeks, mice were sacrificed by cervical dislocation and tibialis anterior muscle was excised from both legs of each mouse. Muscle from one leg was placed in skinning solution before cryopreservation and storage at -80 °C for later analysis through nanoindentation. Muscle from the contralateral leg was snap frozen in liquid nitrogen for western blot analysis.

-Your manuscript must include a complete Additional Information section

We have compiled and written appropriate sections into an additional information section, which includes the following,

Additional information

Data availability statement

Individual datapoints ($n \leq 30$) are included in the figures. Data from the study will be made available upon reasonable request.

Competing interests

The authors have declared that no competing interests exist.

Author Contributions

MJS and JO contributed to the conception of the work. EB, JAR, MJS, and JO designed experiments. EB, JAR, AH, YH, and DGSW performed experiments. EB, AH, DGSW, and TI completed the formal analysis of the data. EB, JAR, AH, DGSW, TI, JO, and MJS interpreted the data. EB completed data visualisation. RDP, MK, JNP, GLC, NRL, SDRH and JO recruited human participants and collected human muscle biopsy samples. EB and MJS wrote the first draft of the manuscript. EB, JAR, AH, RDP, MK, GME, NRL, TI, SDRH, JO and MJS contributed to the manuscript and revised it critically. EB, JAR, AH, YH, DGSW, YL, RDP, MK, JNP, GLC, GME, NRL, TI, SDRH, JO and MJS approved the final version of the manuscript to be published. EB, JAR, AH, DGSW, YH, YL, RDP, MK, JNP, GLC, GME, NRL, TI, SDRH, JO and MJS agreed on all aspects of the work.

Funding

Julien Ochala and Edmund Battey are funded by the Medical Research Council of the UK (MR/S023593/1). Matthew J. Stroud is supported by British Heart Foundation Intermediate Fellowship: FS/17/57/32934 and King's BHF Centre for Excellence Award: RE/18/2/34213. Stephen D. R. Harridge, Norman R. Lazarus and Ross D. Pollock were funded by the Bupa Foundation. Michaeljohn Kalakoutis was funded by a King's College London PhD studentship.

-Please upload separate high-quality figure files via the submission form.

We have submitted high-quality figure files (pdf format) via the submission form.

-You must upload original, uncropped western blot/gel images (including controls) if they are not included in the manuscript. This is to confirm that no inappropriate, unethical or misleading image manipulation has occurred <https://physoc.onlinelibrary.wiley.com/hub/journal-policies#imagmanip> These should be uploaded as 'Supporting information for review process only'. Please label/highlight the original gels so that we can clearly see which sections/lanes have been used in the manuscript figures.

We have uploaded original, uncropped western blot/gel images (including controls) to 'Supporting information for review process only'.

-Please ensure that any tables are in Word format and are, wherever possible, embedded in

the article file itself.

We have ensured that tables are in Word format and are embedded within the article file.

-Please ensure that the Article File you upload is a Word file.

We have uploaded the article as a word file.

-Your paper contains Supporting Information of a type that we no longer publish. Any information essential to an understanding of the paper must be included as part of the main manuscript and figures. The only Supporting Information that we publish are video and audio, 3D structures, program codes and large data files. Your revised paper will be returned to you if it does not adhere to our Supporting Information Guidelines

Because the supplementary figure was not of a kind that the journal publishes, we have removed it from the manuscript.

-A Statistical Summary Document, summarising the statistics presented in the manuscript, is required upon revision. It must be on the Journal's template, which can be downloaded from the link in the Statistical Summary Document section here: https://jp.msubmit.net/cgi-bin/main.plex?form_type=display_requirements#statistics

-Papers must comply with the Statistics Policy https://jp.msubmit.net/cgi-bin/main.plex?form_type=display_requirements#statistics

In summary:

-If $n \leq 30$, all data points must be plotted in the figure in a way that reveals their range and distribution. A bar graph with data points overlaid, a box and whisker plot or a violin plot (preferably with data points included) are acceptable formats.

-If $n > 30$, then the entire raw dataset must be made available either as supporting information, or hosted on a not-for-profit repository e.g. FigShare, with access details provided in the manuscript.

- 'n' clearly defined (e.g. x cells from y slices in z animals) in the Methods. Authors should be mindful of pseudoreplication.

-All relevant 'n' values must be clearly stated in the main text, figures and tables, and the Statistical Summary Document (required upon revision)

-The most appropriate summary statistic (e.g. mean or median and standard deviation) must be used. Standard Error of the Mean (SEM) alone is not permitted.

-Exact p values must be stated. Authors must not use 'greater than' or 'less than'. Exact p values must be stated to three significant figures even when 'no statistical significance' is claimed.

-Statistics Summary Document completed appropriately upon revision

We have completed the statistics summary document and have ensured that the manuscript complies with the statistics policy

-Please include an Abstract Figure file, as well as the figure legend text within the main article file. The Abstract Figure is a piece of artwork designed to give readers an immediate understanding of the research and should summarise the main conclusions. If possible, the image should be easily 'readable' from left to right or top to bottom. It should show the physiological relevance of the manuscript so readers can assess the importance and content of its findings. Abstract Figures should not merely recapitulate other figures in the manuscript. Please try to keep the diagram as simple as possible and without superfluous information that may distract from the main conclusion(s). Abstract Figures must be provided by authors no later than the revised manuscript stage and should be uploaded as a separate file during online submission labelled as File Type 'Abstract Figure'. Please ensure that you include the figure legend in the main article file. All Abstract Figures should be created using BioRender. Authors should use The Journal's premium BioRender account to export high-resolution images. Details on how to use and access the premium account are included as part of this email.

We have included an abstract figure file and the abstract figure legend text within the main article file.

EDITOR COMMENTS

Reviewing Editor:

Your work has been evaluated by two experts in the field. Both reviewers found your work to be novel and interesting. One reviewer requested some additional experimentation. While I agree that additional experimentation could provide important information on the consequences of altering nuclear morphology, such experiments would significantly extend the time to publication and could be saved for follow-up investigations. Please respond the reviewer consequences as comprehensively as possible.

We thank the Editor and the Reviewers for their detailed reviews and for their comments and are very appreciative of the consideration to publish our study in a timely manner. We agree that the additional experimentation could have provided further information, but this would be better investigated in follow-up investigations.

REFeree COMMENTS

Referee #1:

Batthey and colleagues examine myonuclear shape and mechanics in skeletal muscle from young and old humans following exercise and provide novel insight on how age and exercise affect skeletal muscle. This work is innovative and physiologically relevant, and provides new insight into the physiologic adaptations to exercise.

We are very grateful for the Reviewer's insightful comments and compliments on our study, particularly highlighting that the work is innovative and provides new insights in physiological adaptation to exercise.

- First, I would like to commend the authors for their detailed analysis and thorough methods.

We appreciate the Reviewer's praise of our analysis and methods.

- The authors mention that the adaptations to exercise are, in part, mediated by mechanotransduction and provide a brief discussion onto how this could affect gene expression. Due to changes in shape and stiffness being central to this manuscript, I am hoping that the authors can expand upon this section in the discussion and provide a little more detail on how changes in stiffness affects mechanotransduction-mediated gene expression.

This is an excellent suggestion. In the original manuscript, we discussed the possible effects of reduced nuclear stiffness in inactive individuals on mechanotransduction-mediated gene expression on page 19 (lines 494-506). To clarify that these potential consequences may be due to reduced stiffness, we have made the following changes to the paragraph:

- Replaced "mechanical properties" with "greater deformability" (line 494)
- Added "in more compliant myonuclei" (line 498)

"The elongated shape and greater deformability of myonuclei in untrained individuals were reminiscent of those in muscle fibres from humans and mice with muscular dystrophies characterised by muscle wasting and dysfunction (Earle et al., 2020; Tan et al., 2015). Thus, defective myonuclear structure and function due to inactivity may contribute to age-related muscle dysfunction. Specifically, chromatin stretching in more compliant myonuclei may result in expression of genes that contribute to muscle atrophy, which are elevated after two weeks of inactivity (Jones et al., 2004). Additionally, altered chromatin organisation may repress genes encoding contractile or mitochondrial proteins, decreasing force production and endurance capacity (Figure 6B). Deformable myonuclei in untrained individuals may be more susceptible to nuclear envelope rupture, impacting cell health (Earle et al., 2020; Kalukula et al., 2022). These possible consequences of myonuclear dysfunction in old age may collectively contribute to impairments in muscle mass, strength, and endurance with age, and be alleviated by exercise-mediated myonuclear remodelling (Figure 6B)"

Additionally, we have added the following to provide more detail on how changes in nuclear shape, the nuclear lamina, and stiffness affects mechanotransduction-mediated gene expression:

(Page 18-19, lines 475-488):

“More spherical, stiffer myonuclei with increased Lamin A expression in trained individuals may have enhanced transduction of forces via the nuclear lamina, which tethers chromatin (Schreiner et al., 2015). This may directly regulate the expression of genes important for exercise adaptations, by stretching or compacting chromatin to alter transcription factor accessibility (Tajik et al., 2016). Additionally, increased nuclear stiffness and altered lamina composition may affect translocation of transcription factors into the nucleus, given the responsiveness of nuclear pore complexes to cellular forces (Sapra et al., 2020; Zimmerli et al., 2021). For example, greater Lamin A expression may increase translocation of transcriptional co-factors Yes-associated protein/Transcriptional coactivator with PDZ-binding motif (Yap/Taz), which are associated with muscle growth and signalling pathways involved in both endurance and resistance exercise adaptations (Gabriel et al., 2016). Furthermore, altered Lamin A levels may affect nucleo-cytoskeletal shuttling of mechanosensitive transcription factor MRTF-A as reported in Lamin A/C-null mouse embryonic fibroblasts (Ho et al., 2013).”

- It would be interesting to hear the authors thoughts on how changes in stiffness and myonuclear size with aging could contribute to anabolic resistance during aging.

This is an interesting question on the effects of changes in nuclear structure and stiffness on anabolic resistance during ageing. Whilst we do not explicitly refer to how myonuclear stiffness and shape may affect anabolic resistance during ageing in the manuscript, we do consider the potential effects of myonuclear elongation and increased deformability of myonuclei on the expression of atrophic genes and contribution to age-related declines in muscle mass and function (page 24, lines 503-515). These potential changes may attenuate anabolic resistance in master athletes. However, we currently do not have the data to support this and further research would be required to pursue this in these subjects.

- For some of the outcomes (i.e., Figure 1C, Figure 5C) the authors normalized data relative to the sarcomere length. Although the methods are quite detailed, which is appreciated by this reviewer, I am not entirely certain how sarcomere length was quantified (method, software used, number counted per muscle fiber, etc.). It would be good to have a little more detail on this in the methods.

Thank you for pointing out that the method of quantifying sarcomere length was not described in the manuscript. In line with the reviewer’s suggestions, we have included the following description of sarcomere length measurements in the methods section:

(Page 9, lines 213-214)

Sarcomere length was quantified by measuring the distance between ten sarcomeres using the segmented line tool in Fiji and dividing this value by ten.

Additionally, we have added the following description of the experimental procedures involved in determining sarcomere length:

(Page 10, lines 241-242)

Muscle fibres from OT and OU were stretched to different tensions, fixed, and stained with Hoechst and α -Actinin to visualise myonuclei and Z-discs of sarcomeres, respectively.

- The authors should add detail on their animal procedures (i.e., source of animals, access to food and water, temperature, etc.).

We have added the following detail on the animal procedures:

(Page 10, lines 261-270)

Animal handling

Twenty-four male ten-week-old C57BL/6 mice were maintained in groups of four in cages, lined with wood shavings, cardboard rolls and cleaned weekly, in an automated room for photoperiod control (light-dark cycle 12 h/ 12 h). Animals were provided with water and a standard chow diet ad libitum. 12 mice were trained on treadmill over 8 weeks, with a complementary sedentary group left in cages for an equivalent time-period. At the end of the 8 weeks, mice were sacrificed by cervical dislocation and tibialis anterior muscle was excised from both legs of each mouse. Muscle from one leg was placed in skinning solution before cryopreservation and storage at -80 °C for later analysis using nanoindentation. Muscle from the contralateral leg was snap frozen in liquid nitrogen for western blot analysis.

- A recent manuscript by Rader et al.

(<https://physoc.onlinelibrary.wiley.com/doi/full/10.14814/phy2.15476>) appears relevant to this work and should be added to the citations.

We thank the reviewer for drawing our attention to this recent relevant study, and have cited it on page 14 (lines 355),

...myonuclei were strikingly rounder in shape in both younger and older trained individuals (Figure 1A), consistent with recent reports in rodents (Murach et al., 2020; Rader & Baker., 2022).

Referee #2:

In this manuscript, Battey et al. assess the morphological and functional differences between myonuclei from trained and untrained muscle in young and aged individuals. The first portion of the manuscript shows that untrained individuals possess less round/spherical myonuclei when compared to muscle obtained from trained patients. These changes are consistent in both young and aged muscle. By staining isolated fibers for lamin A, the authors also observe a decrease in lamin A in old untrained individuals when compared to their trained counterparts. They deepen this analysis of LINC complex protein alterations by showing increased expression of lamin A and SUN2 in trained mice. Finally, they perform functional assays to show that trained myonuclei from aged individuals are stiffer and less deformable.

The conclusions of the study are supported by the evidence provided. The manuscript follows a clear and concise demonstration, and the text is well written. The data displayed is of high quality and the efforts put to rely on human samples for the majority of the work is impressive and will be appreciated by the community. The observations made could be influential in the fields of exercise physiology and muscle disorders associated with nuclear morphology defects. This does leave the readers awaiting a bit more such as an assessment of DNA damage or nuclear envelope rupture, alterations in gene expression or mechanotransduction signalling. The authors do discuss some of these potential consequences, but experimental explorations would substantially heighten the scope of the manuscript. Overall, this study is well performed and relevant to the muscle field.

We thank the Reviewer for their positive comments and have responded to their thoughtful insights on the potential effects of exercise-induced myonuclear remodelling on DNA damage and nuclear envelope rupture below. We have also discussed more extensively the potential effects of altered myonuclear shape, Lamin A expression, nuclear stiffness and mechanotransduction signalling in the manuscript.

Major points:

As alluded to in the general comments, the authors could strengthen their work by demonstrating the consequences of altered nuclear morphology and lower lamin A levels in untrained myonuclei (DNA damage, nuclear membrane instability, gene expression, etc...). The authors do highlight limitations in human sample quantity thereby hampering a timely investigation of these morphological changes. Nevertheless, this could be performed in mouse as trained mice display increased lamin A and SUN2 expression as well as higher nuclear stiffness. It is unclear if the morphology of mice myonuclei is similarly altered than what is observed in humans but a few key experiments could be done in mice to show potential effects. For example if one session of exercise leads to higher nuclear damage in untrained animals.

We thank the Reviewer for this suggestion and agree that the data presented raise important questions about the potential consequences of altered nuclear morphology and lower Lamin A levels in myonuclei from untrained individuals. As the Reviewer mentioned, the effects of altered myonuclear shape and deformability on DNA damage and nuclear envelope rupture, and alterations in gene expression or mechanotransduction signalling are pertinent future avenues of research to investigate. Interestingly, it has previously been

reported that exercise attenuates DNA damage and increases DNA repair in aged rat skeletal muscle (Radák et al., 2002).

To study this in our system, we have performed WBs to investigate γ H2AX levels (a marker of double-stranded breaks in DNA) in gastrocnemius tissue from trained and untrained mice, but found no significant differences (see Figure below). This was most likely because we collected our samples 3 days after the last exercise session to avoid potentially confounding effects of muscle damage on our primary endpoint measures (Carmichael et al., 2005; Neubauer et al., 2014). After 3 days, the acute effects of exercise on DNA damage would likely no longer be observed (Williamson et al., 2020). To thoroughly investigate the acute effects of exercise on DNA damage in trained and untrained muscle (with differences in stiffness and Lamin A expression), an additional 8-week exercise training study followed by immediate excision of muscle samples would be required. In agreement with the Editor, we feel that this would be beyond the scope of the current manuscript and should be pursued in future studies.

Figure: DNA damage levels using γ H2AX were assessed using Western blot using gastrocnemius muscle from sedentary and exercise mice. Note that no changes were observed.

Minor points:

Figure1:

- The plots are very informative and aesthetic. Could you detail if the colored symbols represent individual means from a single myofiber or from several myofibers pooled in the same experiment.

Thank you for the positive comment on the graphs. The coloured symbols represent the mean values for each individual, calculated from several pooled muscle fibres. This has now been added to the figure legend.

- Panel C shows the normalization of myonuclear aspect ratio over sarcomere length to show no change in trend when taking that factor into account. Can the authors comment if the sarcomere length alone is different in the various cohorts?

Thank you bringing up this important point that helps to communicate further insights into the sarcomere length data. Sarcomere length was $2.0 \pm 0.2 \mu\text{m}$, $2.1 \pm 0.2 \mu\text{m}$, $2.0 \pm 0.07 \mu\text{m}$, and $2.2 \pm 0.4 \mu\text{m}$ in YU, OU, YT, and OT, respectively. We have added this to the results section on page 14 (lines 361-362).

Figure 2:

- The 3D aspect ratio is not as efficiently defined in 3D as it is in 2D. Can the authors better describe the 3D aspect ratio?

Thank you for pointing the lack of clarity in the description of 3D aspect ratio. We have added the following in the methods section to give a clearer definition of 3D aspect ratio (lines 219-223, page 9):

“To quantify three-dimensional shape parameters (sphericity and skeletal length/diameter, referred to as 3D aspect ratio), the DAPI signal was thresholded and analysed using Volocity software. Skeletal length is the maximum length of the object. The selection is eroded evenly from its border inwards until it consists of a one-voxel thick, skeletal representation along its entire length. This "skeleton" is then measured. Skeletal diameter is the diameter of a cylinder if it had a length equal to the skeletal length of the object and a volume equal to the object's measured volume (from Volocity User Guide).”

- Can the authors provide a 3D representative image (similar to Fig2.A) of trained and untrained isolated fibers for comparison.

Whilst we agree that adding a 3D representative image of trained and untrained isolated fibres would aid visual appeal, unfortunately we no longer have access to the software for 3D rendering. Using other imaging software that is available would produce inferior results, which would not be appropriate for the figure in the main text, and adding such images to the supplementary information is not in line with the journal's policy. We hope that the 2D representative images are sufficient for comparison.

- The panels are mis referenced in the text which has made the understanding of this figure more difficult.

Thank you for pointing out this oversight, the figure is now referenced to correctly in the text.

Figure 3:

- The sentence "Thus, to investigate whether exercise affects the organisation of lamin A..." seems grammatically off. Adding line numbers would also help the revision process.

Thank you for pointing the grammatical error in this sentence and the omission of line numbers. We have added line numbers to the manuscript and rephrased the sentence so that it now reads (page 15, lines 385-387),

“Thus, to investigate whether exercise affects the organisation of Lamin A in skeletal muscle, muscle fibres from YU, YT, OU and OT individuals were stained with a Lamin A-specific antibody (Figure 3).”

Figure 5:

- A quick description of a nanoindenter within the text would help the reader understand the assay.

We have amended the text to give a description of the nanoindentation assay (page 16, lines 436-439):

“To confirm this, we used a nanoindenter (which precisely measures mechanical properties of small samples and cells) to physically probe nuclei and directly test the effects of exercise on myonuclear mechanics (Figure 5D-E).”

END OF COMMENTS

References

- Carmichael, M. D., Davis, J. M., Murphy, E. A., Brown, A. S., Carson, J. A., Mayer, E., & Ghaffar, A. (2005). Recovery of running performance following muscle-damaging exercise: Relationship to brain IL-1 β . *Brain, Behavior, and Immunity*, *19*(5), 445–452. <https://doi.org/10.1016/j.bbi.2005.03.012>
- Neubauer, O., Sabapathy, S., Ashton, K. J., Desbrow, B., Peake, J. M., Lazarus, R., Wessner, B., Cameron-Smith, D., Wagner, K. H., Haseler, L. J., & Bulmer, A. C. (2014). Time course-dependent changes in the transcriptome of human skeletal muscle during recovery from endurance exercise: From inflammation to adaptive remodeling. *Journal of Applied Physiology*, *116*(3), 274–287. <https://doi.org/10.1152/jappphysiol.00909.2013>
- Raduk, Z., Naito, H., Kaneko, T., Tahara, S., Nakamoto, H., Takahashi, R., Cardozo-Pelaez, F., & Goto, S. (2002). Exercise training decreases DNA damage and increases DNA repair and resistance against oxidative stress of proteins in aged rat skeletal muscle. *Pflügers Arch-Eur J Physiol*, *445*, 273–278. <https://doi.org/10.1007/s00424-002-0918-6>
- Williamson, J., Hughes, C. M., Burke, G., & Davison, G. W. (2020). A combined γ -H2AX and 53BP1 approach to determine the DNA damage-repair response to exercise in hypoxia. *Free Radical Biology and Medicine*, *154*, 9–17. <https://doi.org/10.1016/j.freeradbiomed.2020.04.026>

Dear Dr Stroud,

Re: JP-RP-2022-284128R1 "Myonuclear alterations associated with exercise are independent of age in humans" by Edmund Battey, Jacob A Ross, Anthony Hoang, Darren G S Wilson, Yu Han, Yotam Levy, Ross D Pollock, Michaeljohn Kalakoutis, Jamie Pugh, Graeme L Close, Georgina M Ellison-Hughes, Norman R Lazarus, Thomas Iskratsch, Stephen D. R. Harridge, Julien Ochala, and Matthew J Stroud

We are pleased to tell you that your paper has been accepted for publication in The Journal of Physiology.

Authors should note that it is too late at this point to offer corrections prior to proofing. The accepted version will be published online, ahead of the copy edited and typeset version being made available. Major corrections at proof stage, such as changes to figures, will be referred to the Editors for approval before they can be incorporated. Only minor changes, such as to style and consistency, should be made at proof stage. Changes that need to be made after proof stage will usually require a formal correction notice.

Yours sincerely,

Michael C. Hogan
Senior Editor
The Journal of Physiology
<https://jp.msubmit.net>
<http://jp.physoc.org>
The Physiological Society
Hodgkin Huxley House
30 Farringdon Lane
London, EC1R 3AW
UK
<http://www.physoc.org>
<http://journals.physoc.org>

P.S. - You can help your research get the attention it deserves! Check out Wiley's free Promotion Guide for best-practice recommendations for promoting your work at www.wileyauthors.com/eeo/guide. You can learn more about Wiley Editing Services which offers professional video, design, and writing services to create shareable video abstracts, infographics, conference posters, lay summaries, and research news stories for your research at www.wileyauthors.com/eeo/promotion.

IMPORTANT NOTICE ABOUT OPEN ACCESS: To assist authors whose funding agencies mandate public access to published research findings sooner than 12 months after publication, The Journal of Physiology allows authors to pay an Open Access (OA) fee to have their papers made freely available immediately on publication.

You can check if your funder or institution has a Wiley Open Access Account here: <https://authorservices.wiley.com/author-resources/Journal-Authors/licensing-and-open-access/open-access/author-compliance-tool.html>.

EDITOR COMMENTS

Reviewing Editor:

Congratulations on a nice article.

REFEREE COMMENTS

Referee #1:

Thank you for sufficiently addressing all of my concerns.

Referee #2:

In this manuscript, Battey et al. assess the morphological and functional differences between myonuclei from trained and untrained muscle in young and aged individuals. The first portion of the manuscript shows that untrained individuals possess less round/spherical myonuclei when compared to muscle obtained from trained patients. These changes are consistent in both young and aged muscle. By staining isolated fibers for lamin A, the authors also observe a decrease in lamin A in old untrained individuals when compared to their trained counter parts. They deepen this analysis of LINC complex proteins alterations by showing increased expression of lamin A and SUN2 in trained mice. Finally, they perform functional assays to show that trained myonuclei from aged individuals are stiffer and less deformable.

All the points were addressed by the authors. Thank you and congratulations on the results.

1st Confidential Review

06-Dec-2022

In this manuscript, Battey et al. assess the morphological and functional differences between myonuclei from trained and untrained muscle in young and aged individuals. The first portion of the manuscript shows that untrained individuals possess less round/spherical myonuclei when compared to muscle obtained from trained patients. These changes are consistent in both young and aged muscle. By staining isolated fibers for lamin A, the authors also observe a decrease in lamin A in old untrained individuals when compared to their trained counter parts. They deepen this analysis of LINC complex proteins alterations by showing increased expression of lamin A and SUN2 in trained mice. Finally, they perform functional assays to show that trained myonuclei from aged individuals are stiffer and less deformable.

All the points were addressed by the authors. Thank you and congratulations on the results.